# Sporadic Dissemination of *tet*(X3) and *tet*(X6) Mediated by Highly Diverse Plasmidomes among Livestock-Associated *Acinetobacter*

Ying-Ying Cheng,[a,b,c] Yang Liu,[a,b] Yong Chen,[a,b] Fu-Man Huang,[d] Rong-Chang Chen,[a,b] Yong-Hong Xiao,[e] Kai Zhou[a,b]

aShenzhen Institute of Respiratory Diseases, Second Clinical Medical College (Shenzhen People's Hospital), Jinan University, Guangzhou, China

bThe First Affiliated Hospital (Shenzhen People's Hospital), Southern University of Science and Technology, Shenzhen, China

cForensics Genomics International (FGI), BGI-Shenzhen, Shenzhen, China

dCollege of Biotechnology, Guilin Medical University, Guilin, China

eCollaborative Innovation Center for Diagnosis and Treatment of Infectious Diseases, State Key Laboratory for Diagnosis and Treatment of Infectious Diseases, The First Affiliated Hospital, College of Medicine, Zhejiang University, Hangzhou, China

Ying-Ying Cheng and Yang Liu contributed equally to this article. The author order was determined by their equal but gradated contributions for this article.

**ABSTRACT** The emergence of high-level tigecycline resistance mediated by plasmid-borne *tet*(X) genes greatly threatens the clinical effectiveness of tigecycline. However, the dissemination pattern of plasmid-borne *tet*(X) genes remains unclear. We here recovered *tet*(X)-positive *Acinetobacter* isolates from 684 fecal and environmental samples collected at six livestock farms. Fifteen *tet*(X)-positive *Acinetobacter* isolates were identified, mainly including 9 *tet*(X3)- and 5 *tet*(X6)-positive *Acinetobacter towneri* isolates. A clonal dissemination of *tet*(X3)-positive *A. towneri* was detected in a swine farm, while the *tet*(X6)-positive *A. towneri* isolates mainly disseminated sporadically in the same farm. A *tet*(X3)-carrying plasmid (pAT181) was self-transmissible from a tigecycline-susceptible *A. towneri* strain to *Acinetobacter baumannii* strain ATCC 17978, causing 64- to 512-fold increases in the MIC values of tetracyclines (including tigecycline). Worrisomely, pAT181 was stably maintained and increased the growth rate of strain ATCC 17978. Further identification of *tet*(X) genes in 10,680 *Acinetobacter* genomes retrieved from GenBank revealed that *tet*(X3) (*n* = 249), *tet*(X5)-like (*n* = 61), and *tet*(X6) (*n* = 53) were the prevalent alleles mainly carried by four species, and most of them were livestock associated. Phylogenetic analysis showed that most of the *tet*(X3)- and *tet*(X6)-positive isolates disseminated sporadically. The structures of the *tet*(X3), and *tet*(X6) plasmidomes were highly diverse, and no epidemic plasmids were detected. However, cross-species and cross-region transmissions of *tet*(X3) might have been mediated by several plasmids in a small proportion of strains. Our study implies that horizontal plasmid transfer may be insignificant for the current dissemination of *tet*(X3) and *tet*(X6) in *Acinetobacter* strains. Continuous surveillance for *tet*(X) genes in the context of One Health is necessary to prevent them from transmitting to humans.

**IMPORTANCE** Recently identified plasmid-borne *tet*(X) genes have greatly challenged the efficiency of tigecycline, a last-resort antibiotic for severe infection, while the dissemination pattern of the plasmid-borne *tet*(X) genes remains unclear. In this study, we identified a clonal dissemination of *tet*(X3)-positive *A. towneri* isolates on a swine farm, while the *tet*(X6)-positive *A. towneri* strains mainly disseminated sporadically on the same farm. Of more concern, a *tet*(X3)-carrying plasmid was found to be self-transmissible, resulting in enhanced tigecycline resistance and growth rate of the recipient. Further exploration of a global data set of *tet*(X)-positive *Acinetobacter* genomes retrieved from GenBank revealed that most of the *tet*(X3)- and *tet*(X6)-positive isolates shared a highly distant relationship, and the structures of *tet*(X3) and *tet*(X6) plasmidomes exhibited high mosaicism. Notably, some of the isolates belong to

Address correspondence to Kai Zhou, Kai_Zhou@zju.edu.cn.

*Acinetobacter* species that are opportunistic pathogens and have been identified as sources of nosocomial infections, raising concerns about transmission to humans in the future. Our study evidenced the sporadic dissemination of *tet*(X3) and *tet*(X6) in *Acinetobacter* strains and the necessity of continuous surveillance for *tet*(X) genes in the context of One Health.

**KEYWORDS** plasmid-borne tigecycline resistance, *tet*(X3), *tet*(X6), *Acinetobacter*, self-transmissible plasmid

Tigecycline is used to treat a wide range of clinical infections caused by Gram-positive and Gram-negative bacteria with multidrug resistance (MDR). With the global dissemination of carbapenemases and mobilized colistin resistance (*mcr*) genes in recent years, this broad-spectrum tetracycline family antibiotic has been raised to be a last-line treatment regimen in clinical settings (1–6). However, the increasing occurrence of transferable tigecycline inactivation genes [*tet*(X) genes] is threatening the clinical efficacy of tigecycline (7, 8).

The first flavin-dependent monooxygenase gene, named *tet*(X), was identified in Tn*4351* and Tn*4400* on the chromosome of *Bacteroides fragilis* in 1990 (9). Subsequently, 14 chromosome-carried and plasmid-mediated *tet*(X) genes, *tet*(X1) to *tet*(X14), have been reported in various species originating from animals, humans, and the environment (10–12). These Tet(X) enzymes, except for Tet(X1), exhibited different levels of activity against almost all tetracyclines, including a new tetracycline, eravacycline, that was approved by the U.S. Food and Drug Administration (FDA) in 2018 (4, 12, 13). The first plasmid-borne *tet*(X3) and *tet*(X4) genes were found in livestock-associated *Acinetobacter baumannii* and *Escherichia coli* strains, respectively, in 2019 (7), raising the concern of horizontal transfer of tigecycline resistance. Since then, additional *tet*(X) alleles have been reported to be plasmid borne, including *tet*(X5) and *tet*(X6) and their variants. Epidemiological studies reveal that these novel *tet*(X) orthologs have mainly circulated in animals in China due to the heavy use of tetracyclines in husbandry (8). In some pioneering studies, IS*CR2* was highlighted as the key element facilitating the horizontal transfer of *tet*(X) genes, through circular intermediates (14–17). However, the role of mobile elements in the dissemination of *tet*(X) genes remains obscure.

The *tet*(X) genes have been detected in over 16 bacterial species, and *Acinetobacter* spp. were among the major hosts (7, 11, 17–20). Currently, most of the *tet*(X)-positive *Acinetobacter* species isolates have been associated with livestock, and very few have been found in humans (16, 21). A surveillance study at avian farms in China showed that 1.6% to 18.3% of *Acinetobacter* species strains were *tet*(X) positive (22). Another surveillance study for tigecycline-resistant *Acinetobacter* spp. from 2015 to 2018 in 14 provinces and municipalities in China identified 2.3% to 25.3% *tet*(X)-positive isolates from pig farms, migratory birds, and samples from human (20). Plasmid-borne and/or chromosome-carried *tet*(X3) and *tet*(X6) were prevalent in livestock-associated *Acinetobacter* species isolates, and *tet*(X5) has so far only been detected in an *A. baumannii* strain from humans (7, 16, 20, 22, 23).

In this study, surveillance of *tet*(X)-positive *Acinetobacter* species isolates recovered from livestock and their surrounding environmental sources was performed at six livestock farms located in Zhejiang Province in 2019. The epidemiological and genetic characterizations of *tet*(X)-positive isolates and *tet*(X)-harboring plasmids were dissected. We further investigated the population structure and distribution of *tet*(X)-positive *Acinetobacter* strains identified in a public database, as well as the plasmidomes of *tet*(X3) and *tet*(X6).

## RESULTS

***A. towneri* was the prevalent species carrying *tet*(X) genes among *Acinetobacter* strains collected in this study.** Two hundred ninety-two isolates were recovered from 534 stool samples and 150 environmental samples collected from 2 swine farms, 2 dairy farms, and 2 sheep farms, including 215 isolates of *Acinetobacter* spp. and 77

**TABLE 1** *tet*(X)-positive strains isolated in this study

| Strain | Species | Gene | Location | Source | Sequencing platform | Genome accession no. |
|---|---|---|---|---|---|---|
| ZJ202 | *Empedobacter stercoris* | *tet*(X2) | Chromosome | Fecal, swine farm 1 | Illumina | JABFOQ000000000 |
| ZJ180 | *E. stercoris* | *tet*(X2) | Chromosome | Fecal, swine farm 1 | Illumina | JACXZB000000000 |
| ZJ215 | *E. stercoris* | *tet*(X2) | Chromosome | Fecal, swine farm 1 | Illumina | JACXZC000000000 |
| ZJ286 | *Myroides odoratimimus* | *tet*(X2) | NA[a] | Soil, swine farm 2 | Illumina | JACXZD000000000 |
| ZJ291 | *M. odoratimimus* | *tet*(X2) | NA | Soil, swine farm 2 | Illumina | JACXZE000000000 |
| ZJ295 | *M. odoratimimus* | *tet*(X2) | NA | Soil, swine farm 2 | Illumina | JACXZF000000000 |
| AT184 | *Acinetobacter towneri* | *tet*(X3) | Plasmid | Fecal, swine farm 1 | Nanopore | JACXZG000000000 |
| ZJ199 | *Acinetobacter* sp. | *tet*(X3) | Chromosome | Fecal, swine farm 1 | Nanopore | CP062182 |
| AT200 | *A. towneri* | *tet*(X3) | Plasmid | Fecal, swine farm 1 | Illumina | JACXZH000000000 |
| AT216 | *A. towneri* | *tet*(X3) | Plasmid | Fecal, swine farm 1 | Illumina | JACXZI000000000 |
| AT217 | *A. towneri* | *tet*(X3) | Plasmid | Fecal, swine farm 1 | Illumina | JACXZJ000000000 |
| AT181 | *A. towneri* | *tet*(X3) | Plasmid | Fecal, swine farm 1 | Nanopore | JACXZK000000000 |
| AT209 | *A. towneri* | *tet*(X3) | Plasmid | Fecal, swine farm 1 | Illumina | JACXZL000000000 |
| AT211 | *A. towneri* | *tet*(X3) | Plasmid | Fecal, swine farm 1 | Illumina | JACXZM000000000 |
| AT213 | *A. towneri* | *tet*(X3) | Plasmid | Fecal, swine farm 1 | Illumina | JACXZN000000000 |
| AT214 | *A. towneri* | *tet*(X3) | Plasmid | Fecal, swine farm 1 | Illumina | JACXZO000000000 |
| AT185 | *A. towneri* | *tet*(X6), *tet*(X6) | Plasmid | Fecal, swine farm 1 | Illumina | JACXZP000000000 |
| AT208 | *A. towneri* | *tet*(X6) | Plasmid | Fecal, swine farm 1 | Illumina | JACXZQ000000000 |
| AT232 | *A. towneri* | *tet*(X6) | Plasmid | Fecal, swine farm 1 | Nanopore | CP062183-CP062184 |
| AT235 | *A. towneri* | *tet*(X6) | Plasmid | Fecal, swine farm 1 | Nanopore | CP062185-CP062186 |
| AT205 | *A. towneri* | *tet*(X6) | Plasmid | Fecal, swine farm 1 | Nanopore | CP048014-CP048018 |
| ZJ183 | *E. stercoris* | *tet*(X14), *tet*(X2), *tet*(X2) | Chromosome | Fecal, swine farm 1 | Nanopore | CP053698-CP053701 |
| ZJ182 | *E. stercoris* | *tet*(X14)-tet(X2) | Chromosome | Fecal, swine farm 1 | Illumina | JACXZR000000000 |

[a]NA, not available: the location of *tet*(X) gene cannot be resolved in this genome.

isolates belonging to other species. Twenty-three *tet*(X)-positive isolates were identified (7.88%; 23/292), including 15 *Acinetobacter* species isolates (6.97%; 15/215), 5 *Empedobacter stercoris* isolates, and 3 *Myroides odoratimimus* isolates (Table 1). The 23 *tet*(X)-positive isolates were exclusively isolated from swine farms. The *Acinetobacter* spp. and *E. stercoris* isolates were all recovered from the fecal samples of swine farm 1, and the 3 *M. odoratimimus* isolates were from the soil samples of swine farm 2.

The 15 *tet*(X)-positive *Acinetobacter* species isolates were assigned by average nucleotide identity (ANI) analysis to *Acinetobacter towneri* (*n* = 14) and an unclassified species (*n* = 1), and the other 8 *tet*(X)-positive isolates were *E. stercoris* (*n* = 5) and *M. odoratimimus* (*n* = 3) (Table 1). Four different *tet*(X) genes [*tet*(X2), *tet*(X3), *tet*(X6), and *tet*(X14)] were identified in the 23 *tet*(X)-positive isolates (Table 1). *tet*(X2) was exclusively detected in the 8 non-*Acinetobacter* isolates, and *tet*(X3) was in 9 *A. towneri* isolates and 1 unclassified species isolate (ZJ199). *tet*(X6) and *tet*(X14) were found in 5 *A. towneri* and 2 *E. stercoris* isolates, respectively. Notably, two copies of *tet*(X6) were carried by an *A. towneri* isolate (AT185). Eight of the *tet*(X3)-positive *Acinetobacter* species isolates clustered together, with 3 to 36 single-nucleotide polymorphisms (SNPs) (Fig. 1), suggesting the clonal dissemination of one strain. Two of the *tet*(X6)-positive isolates were also clonally related (1 SNP). The remaining five isolates showed distant relationships (26,876 to 31,071 SNPs), indicating sporadic dissemination of these strains.

**Antimicrobial resistance profiles of *tet*(X)-carrying isolates.** Eight of the 23 *tet*(X)-positive isolates (34.78%) were resistant to tigecycline, with MIC values at 1 to 2 mg/liter, encompassing 4 *tet*(X3)-positive *A. towneri* isolates, 1 *tet*(X6)-positive *A. towneri* isolate, 2 *tet*(X2)- and *tet*(X14)-positive *E. stercoris* isolates, and 1 *tet*(X2)-positive *M. odoratimimus* isolate (Table 2). Five tigecycline-resistant isolates (3 *A. towneri* isolates and 2 *E. stercoris* isolates) additionally exhibited resistance to the newly FDA-approved eravacycline, with MIC values at 1 to 4 mg/liter. Except for the isolate carrying 2 copies of *tet*(X6), the other 14 *Acinetobacter* species isolates were resistant to tetracycline with MIC values of ≥16 mg/liter (Table 2). Strain AT232 showed significantly higher resistance to tetracyclines than the other 13 isolates, which might be caused by the presence of a two-component system, AdeSR, involved in the expression of the AdeABC efflux pump (24). Four and two *Acinetobacter* species isolates additionally showed resistance to ciprofloxacin

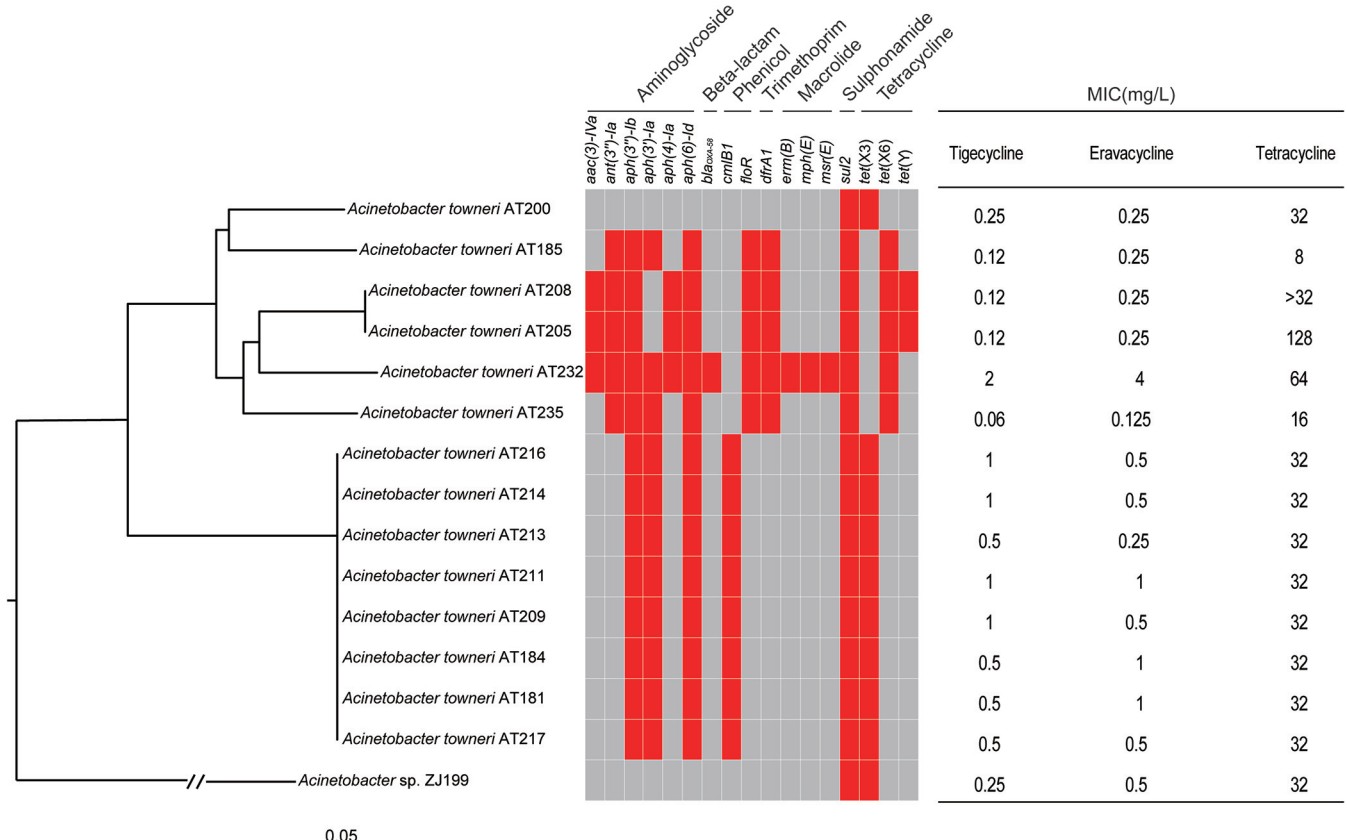

**FIG 1** Phylogenetic analysis of *tet*(X)-positive *Acinetobacter* isolates collected in this study. The core-genome SNPs of *tet*(X)-encoding strains were used to generate the phylogenetic tree. The tree is rooted at strain ZJ199. The ARGs of each strain are exhibited by the heatmap, and the existence of ARGs is in red. MIC values of each strain against tetracyclines are listed. AT205 has been reported previously (26).

and doxycycline, respectively (Table 2). All of the *tet*(X)-positive *Acinetobacter* species isolates were susceptible to colistin and carbapenems. The *M. odoratimimus* isolates were resistant to colistin and carbapenems due to intrinsic resistance (25).

The 23 *tet*(X)-positive isolates were subjected to whole-genome sequencing (WGS) (Table S1 in the supplemental material). All of the *A. towneri* strains were multidrug resistant (MDR), and more antibiotic resistance genes (ARGs) were detected in the *tet*(X6)-carrying clone (mean = 8.67; median = 9) than in the *tet*(X3)-carrying clone (mean = 6; median = 6), albeit the difference was not significant (*P* > 0.05) (Fig. 1). The 8 strains of the *tet*(X3)-carrying clone shared an identical resistome [*aph(3″)-Ib*, *aph(3′)-Ia*, *aph(6)-Id*, *cmlB1*, *sul2*, and *tet*(X3)], further supporting the aforementioned clonal dissemination (Fig. 1), while the resistomes of the *tet*(X6)-carrying strains were highly diverse, with genes that included the following: *aacC4*, *ant(3″)-Ia*, and *aph(4)-Ia*, encoding resistance to aminoglycosides; *bla*<sub>OXA-58</sub>, encoding resistance to beta-lactams; *floR*, encoding resistance to phenicols; *dfrA1*, encoding resistance to trimethoprim; *erm(B)*, *mph(E)*, and *msr(E)*, encoding resistance to macrolides; and *tet*(X6) and *tet*(Y), encoding resistance to tetracyclines (Fig. 1). Only two ARGs were detected in strain ZJ199 [*sul2* and *tet*(X3)]. The resistomes of *E. stercoris* and *M. odoratimimus* were different from those of *Acinetobacter* spp. (Table S2).

**tet(X3) and tet(X6) were harbored by various plasmids.** To understand the vectors of the two prevalent *tet*(X) genes, i.e., *tet*(X3) and *tet*(X6), five representative strains [AT181 and AT184 for *tet*(X3) and ZJ199, AT232, and AT235 for *tet*(X6)] were additionally chosen for long-read sequencing based on their antimicrobial resistance profiles and the genetic environments of their *tet*(X) genes (Table S1). The hybrid assembly

**TABLE 2** MIC values of antibiotics tested in this study

| Strain | CAZ | CRO | FEP | IPM | MEM | CIP | LVX | AMK | GEN | SXT | CSL | COL | TGC | OTC | CTC | DMC | DOX | MIN | ERV | TET |
|---|---|---|---|---|---|---|---|---|---|---|---|---|---|---|---|---|---|---|---|---|
| ZJ202 | 4 | 2 | 0.125 | 0.25 | 0.125 | 1 | 0.5 | 16 | 8 | 0.25 | 2 | 16 | 0.5 | 32 | 4 | 4 | 1 | 0.5 | 0.5 | 16 |
| ZJ180 | 2 | 2 | 0.125 | 0.5 | 0.25 | 1 | 0.5 | 16 | 4 | 0.06 | 4 | 32 | 0.5 | 16 | 4 | 2 | 0.5 | 0.25 | 1 | 8 |
| ZJ215 | 2 | 2 | 0.25 | 0.5 | 0.125 | 0.125 | 0.5 | 2 | 4 | >8 | 0.25 | 16 | 0.5 | 32 | 4 | 4 | 1 | 0.5 | 1 | 16 |
| ZJ286 | 64 | >64 | 8 | >32 | 2 | >32 | 8 | >128 | >128 | 1 | >128 | >32 | 0.5 | >128 | >128 | >128 | >128 | 128 | 1 | >128 |
| ZJ291 | 64 | >64 | 8 | >32 | 2 | >32 | 16 | >128 | >128 | >8 | >128 | >32 | 2 | >128 | >128 | >128 | 64 | 32 | 1 | >128 |
| ZJ295 | 64 | >64 | 8 | >32 | 2 | >32 | 8 | >128 | >128 | 0.5 | >128 | >32 | 0.5 | >128 | >128 | >128 | >128 | 16 | 0.5 | >128 |
| AT184 | 2 | 4 | 0.5 | 0.125 | 0.03 | 1 | 1 | 1 | 1 | >8 | 1 | 0.5 | 0.5 | 128 | 16 | 16 | 1 | 0.5 | 1 | 32 |
| ZJ199 | 0.25 | 0.25 | 0.06 | 0.03 | 0.03 | 4 | 2 | 0.06 | 0.125 | >8 | 0.06 | 1 | 0.25 | 128 | 16 | 8 | 2 | 0.25 | 0.5 | 32 |
| AT200 | 2 | 4 | 0.25 | 0.125 | 0.03 | 0.03 | 0.06 | 0.25 | 0.125 | >8 | 0.5 | 2 | 0.25 | 64 | 8 | 4 | 0.5 | 0.5 | 0.25 | 32 |
| AT216 | 2 | 4 | 0.5 | 0.125 | 0.06 | 2 | 0.5 | 1 | 0.25 | >8 | 0.25 | 1 | 1 | 64 | 16 | 8 | 0.5 | 0.25 | 0.5 | 32 |
| AT217 | 2 | 4 | 0.5 | 0.125 | 0.06 | 2 | 0.5 | 1 | 0.25 | 8 | 0.25 | 1 | 0.5 | 128 | 16 | 16 | 0.5 | 0.25 | 0.5 | 32 |
| AT181 | 2 | 4 | 0.25 | 0.125 | 0.06 | 1 | 0.5 | 1 | 0.5 | >8 | 1 | 1 | 0.5 | 128 | 16 | 16 | 1 | 0.5 | 1 | 32 |
| AT209 | 2 | 4 | 0.25 | 0.125 | 0.03 | 0.03 | 0.5 | 1 | 0.5 | >8 | 1 | 0.5 | 1 | 128 | 16 | 8 | 0.5 | 0.5 | 0.5 | 32 |
| AT211 | 2 | 4 | 0.25 | 0.125 | 0.03 | 0.03 | 0.5 | 1 | 0.5 | >8 | 1 | 1 | 1 | 128 | 16 | 8 | 1 | 0.25 | 1 | 32 |
| AT213 | 2 | 4 | 0.25 | 0.125 | 0.03 | 0.03 | 0.5 | 2 | 0.5 | >8 | 1 | 1 | 0.5 | 128 | 16 | 8 | 0.5 | 0.5 | 0.25 | 32 |
| AT214 | 2 | 4 | 0.25 | 0.125 | 0.03 | 0.03 | 0.5 | 2 | 0.5 | >8 | 1 | 1 | 1 | 64 | 8 | 8 | 0.25 | 0.5 | 0.5 | 32 |
| AT185 | 2 | 4 | 0.5 | 0.25 | 0.03 | 1 | 0.5 | 0.5 | 0.25 | >8 | 1 | 2 | 0.12 | 32 | 8 | 4 | 0.25 | 0.25 | 0.25 | 8 |
| AT208 | 2 | 4 | 0.25 | 0.25 | 0.03 | 0.03 | 1 | 1 | 8 | >8 | 1 | 2 | 0.12 | >128 | 128 | 128 | 16 | 2 | 0.25 | >32 |
| AT232 | 2 | 4 | 0.5 | 0.25 | 0.06 | 4 | 1 | 0.5 | 4 | 8 | 0.5 | 2 | 2 | 128 | 64 | 32 | 4 | 2 | 4 | 64 |
| AT235 | 2 | 4 | 0.5 | 0.125 | 0.03 | 4 | 1 | 0.5 | 0.125 | 8 | 0.25 | 2 | 0.06 | 32 | 4 | 2 | 0.25 | 0.25 | 0.125 | 16 |
| AT205 | 4 | 8 | 0.5 | 0.5 | 0.06 | 4 | 1 | 1 | 8 | >8 | 1 | 2 | 0.12 | 128 | 128 | 128 | 32 | 0.5 | 0.25 | 128 |
| ZJ183 | 2 | 4 | 0.5 | 0.25 | 0.125 | 1 | 1 | 32 | 16 | 0.06 | 4 | 32 | 1 | 128 | 8 | 8 | 4 | 0.125 | 1 | 16 |
| ZJ182 | 1 | 1 | 0.06 | 0.125 | 0.125 | 2 | 1 | 16 | 8 | 0.06 | 2 | 32 | 1 | 64 | 8 | 8 | 2 | 1 | 2 | 16 |

*a*CAZ, ceftazidime; CRO, ceftriaxone; FEP, cefepime; IPM, imipenem; MEM, meropenem; CIP, ciprofloxacin; LVX, levofloxacin; AMK, amikacin; GEN, gentamycin; SXT, sulfamethoxazole-trimethoprim; CSL, cefoperazone-sulbactam; COL, colistin; TGC, tigecycline; OTC, oxytetracycline; CTC, chlortetracycline; DMC, demeclocycline; DOX, doxycycline; MIN, minocycline; ERV, eravacycline; TET, tetracycline.

confirmed that *tet*(X3) and *tet*(X6) were plasmid borne in the four *A. towneri* isolates, and a chromosome-carried *tet*(X3) was detected in ZJ199.

The *tet*(X3)-carrying plasmids detected in AT181 (pAT181) and AT184 (pAT184) were circularized (confirmed by PCR) and identical, with a size of 75,969 bp. These two plasmids were untypeable, with an average GC content of 42.5%. Multiple ARGs were carried by the two plasmids, including *aph(3′)-Ia*, *aph(3″)-Ib*, *aph(6)-Id*, *sul2*, and *tet*(X3). BLAST analysis of the nucleotide sequence of pAT181 in GenBank showed that the best match was a transferable *tet*(X3)-harboring plasmid, p10FS3-1-3 (accession number CP039146) (100% identity and 97% coverage) carried by a novel species of *Acinetobacter* (20). Others sharing a high similarity with pAT181 included a *tet*(X5)-harboring plasmid, pAB17H194-1 (accession number CP040912; 99.95% identity and 86% coverage), carried by an *A. pittii* strain and a *tet*(X3)-harboring plasmid, p18TQ-X3 (accession number CP045132; 99.99% identity and 80% coverage), carried by an *A. indicus* strain. These data suggested that pAT181-like plasmids have disseminated among various species of *Acinetobacter*.

In accordance with the phylogeny, the *tet*(X3)-carrying plasmids carried by the 8 clonal isolates were all homologous to pAT181, with >90% coverage and nucleotide identity (Fig. S1A), and the *tet*(X3)-carrying plasmid carried by AT200 was different from pAT181, with <50% coverage and >90% identity (Fig. S1A). The best match for pAT200 in GenBank was p10FS3-1-3, with 58.77% coverage and 70% identity.

The two *tet*(X6)-harboring circularized plasmids pAT232 and pAT235 shared as little as 38% coverage and 99.95% identity; however, the sequences of their *rep* genes were identical, indicating that they might originate from a common ancestor. pAT232 was 186,508-bp in length, with a GC content of 41.03%. A BLAST search against GenBank showed that the best matches for pAT232 were a *tet*(X6)-carrying plasmid, pAT205 (accession number CP048015) (76% coverage and 99.99% identity), carried by *A. towneri* strain AT205 isolated on the same swine farm (26), and a *tet*(X)-negative plasmid, p19110F47-2 (accession number CP046044) (70% coverage and 99.99% identity), carried by an *A. towneri* strain isolated from pigs. pAT235 was 124,466 bp in length, with a

GC content of 41.16%. The best matches for pAT235 were pAT205 (49% coverage and 100% identity) and a *tet*(X3)-harboring plasmid, pGX7 (accession number CP071772) (44% coverage and 99.95% identity), detected in an *A. towneri* strain isolated from pigs in China. These data suggest that pAT232 and pAT235 might originate from *A. towneri* strains associated with pigs.

When pAT232 was used as a reference to identify the plasmids bearing *tet*(X6) in the other *tet*(X6)-positive isolates collected here, AT208 showed the highest similarity to pAT232 (77.84% coverage and 99.16% identity) (Fig. S1B). When pAT235 was used as a reference, AT185 shared 100% coverage and 94.51% identity (Fig. S1C), suggesting that a pAT235-like *tet*(X6)-encoding plasmid was harbored in AT185. Of note, AT185 was genetically distant from AT235, with 30,097 SNPs (Fig. 1). A pAT205-like *tet*(X6)-harboring plasmid was detected in AT208 when pAT205 was used as a reference (100% coverage and 96.48% identity) (Fig. S1D). These results reveal that horizontal transfers of *tet*(X6)-carrying plasmids might have occurred sporadically.

**Genetic environments of *tet*(X3) and *tet*(X6).** The genetic environments of plasmid-borne *tet*(X3) [ΔIS*CR2*-*xerD*-*tet*(X3)-*res*-IS*CR2*] detected in 8 of 9 *A. towneri* strains were identical and highly similar to that of the prototype detected in *A. baumannii* strain 34AB (Fig. 2A) (7). To fully understand the distribution of this genetic environment among *tet*(X3)-carrying *Acinetobacter* strains, we used BLAST to compare it to 249 *tet*(X3)-carrying *Acinetobacter* genomes retrieved from GenBank (see below). The fragment ΔIS*CR2*-*xerD*-*tet*(X3)-*res*-IS*CR2* was detected on a single contig of 21.3% (53/249) of genomes with >90% coverage and identity. The proportion increased to 86.35% (215/249) when matches on different contigs were counted together, implying a major structure encoding *tet*(X3) in *Acinetobacter* spp. A different *tet*(X3) genetic environment [IS*4*-IS*4*-*tet*(X3)-*res*-ΔIS*CR2*] was detected on the chromosome of strain ZJ199 (Fig. 2A).

The genetic environments of *tet*(X6) were much more diverse than those of *tet*(X3) detected in our collection (Fig. 2B). A 7,270-bp composite structure [ΔIS*CR2*-IS*30*-*tet*(X6)-*abh*-*guaA*-IS*CR2*] was detected in pAT232, which is similar to the prototype [ΔIS*CR2*-*tet*(X6)-*abh*-*guaA*-IS*CR2*] identified in pAT205 and a *Proteus* genomospecies 6 strain (26, 27), except for an insertion of an IS*30*. The *tet*(X6) located within a 6,885-bp region [IS*CR2*-*fabF*-*tet*(X6)-*abh*-*glmM*-*sul2*] in pAT235 (Fig. 2B) was almost identical to that detected on the chromosome of *A. indicus* strain Q186-3_T (100% coverage and 99.58% identity) and on pABF9692, carried by an *A. baumannii* strain (accession number CP048828) (100% coverage and 98.70% identity). In strain AT185, the genetic context of one copy of *tet*(X6) was identical to that detected in pAT235, and a truncated structure was found for the other copy (Fig. 2B). The IS*CR2*-*fabF*-*tet*(X6)-*abh* fragment was also found on the chromosomes of *A. indicus* strain LYS68A (CP070997) and *A. baumannii* strain 31FS3-2 (CP0445177), indicating that this structure might mediate the mobilization of *tet*(X6) between plasmids and chromosomes in *Acinetobacter* spp.

**A *tet*(X3)-carrying plasmid was self-transmissible from *A. towneri* to *A. baumannii* and increased its resistance to tetracyclines and growth rate.** A conjugation assay was performed to test the transferability of *tet*(X)-encoding plasmids. We only obtained tigecycline-resistant *A. baumannii* transconjugants from *A. towneri* strain AT181, with frequencies of $1.85 \times 10^{-6}$ per recipient cell. Multiple attempts at plasmid transfers failed when *E. coli* strain EC600 was used as a recipient. Compared with those of the recipient strain ATCC 17978, the MIC values of tigecycline and the other tetracyclines against the transconjugant ATCC 17978-pAT181 increased by 128-fold and ~64- to 512-fold, respectively (Table S3). WGS was performed for ATCC 17978-pAT181 and ATCC 17978 to detect the transferable structure of *tet*(X3). A unique plasmid, pAT181, was detected in the transconjugant ATCC 17978-pAT181, demonstrating that the transmission of tigecycline resistance was mediated by pAT181 (Fig. S2). This is different from another self-transmissible *tet*(X3)-harboring plasmid p10FS3-1-3 in that the transfer of p10FS3-1-3 into *Acinetobacter baylyi* strain ADP1 did not bring a significant additive effect to the resistance to tetracyclines (20). To the best of our knowledge, this is the first report showing the horizontal transfer of a *tet*(X3)-carrying plasmid conferring tetracycline resistance to the recipient.

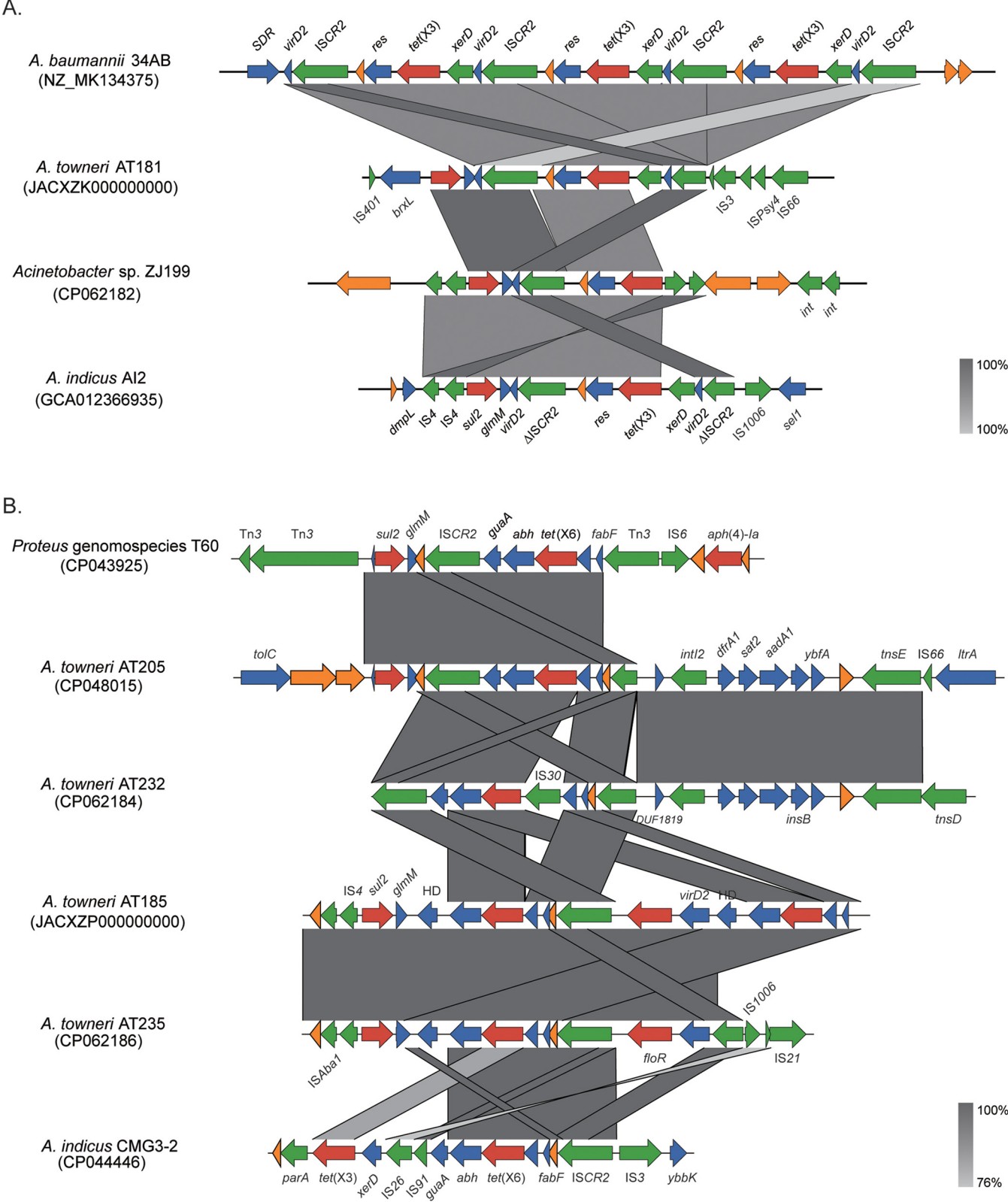

**FIG 2** Genetic context of *tet*(X3) and *tet*(X6) genes identified in *Acinetobacter* spp. (A) Comparison of the genetic contexts of *tet*(X3). The genomic contexts of *tet*(X3) identified in *A. baumannii* strain 34AB (accession number MK134375) and *A. indicus* strain AI2 (accession number GCA_012366935) are used as the reference sequences. (B) Comparison of the genetic contexts of *tet*(X6). The genomic contexts of *tet*(X6) identified in *Proteus* genomospecies 6 T60 (accession number CP043925) and *A. indicus* strain CMG3-2 (accession number CP044446) are used as the reference sequences. Genes are indicated by color-coded arrows dependent on the functional annotations and direction of transcription. ARGs are in red; mobile genetic element genes are in green; genes with other functions are in blue; hypothetical genes are in orange.

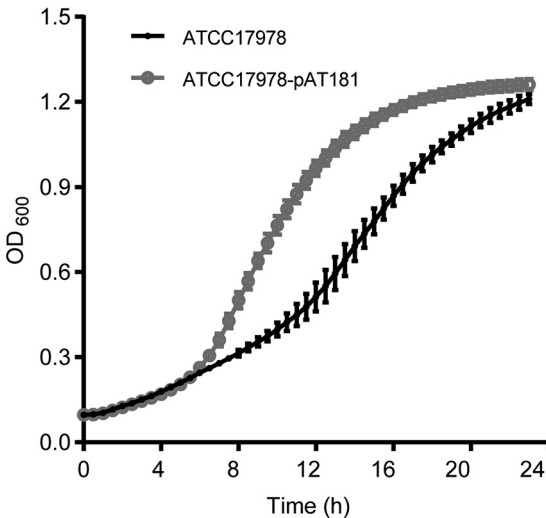

**FIG 3** Growth curves of the recipient strain *A. baumannii* ATCC 17978 and the transconjugant strain ATCC 17978-pAT181 at 37°C. The optical density at 600 nm was recorded every 30 min. The assay was in triplicate.

*tet*(X3) was stable in the recipient strain ATCC 17978 without antibiotic stress during 10 days of passage, with a 100% retention rate. Compared with that of ATCC 17978, the doubling time of ATCC 17978-pAT181 was shortened from 4.59 h to 2.91 h (Fig. 3). These results suggest that pAT181 could facilitate the dissemination of *tet*(X3) among *Acinetobacter* spp. strains.

**_tet_(X3), _tet_(X5)-like, and _tet_(X6) are the prevalent alleles among the _tet_(X) family and disseminate sporadically in four species of _Acinetobacter_ spp.** To fully understand the distribution of *tet*(X) genes among *Acinetobacter* spp., a BLAST comparison of the nucleotide sequences of 15 known *tet*(X) alleles and their variants to 10,680 *Acinetobacter* genomes retrieved from GenBank was performed. *tet*(X3) was found in 249 genomes, *tet*(X4) in 9 genomes, *tet*(X5) ($n = 2$), *tet*(X5.2) ($n = 53$), and *tet*(X5.3) ($n = 6$) in 61 genomes, *tet*(X6) in 53 genomes, and *tet*(X13), a 1-residue variant of *tet*(X6), in 4 genomes. These data reveal that *tet*(X3), *tet*(X5.2), and *tet*(X6) are the prevalent *tet*(X) genes among *Acinetobacter* spp.

Species identification by ANI analysis showed three predominant *Acinetobacter* species carrying *tet*(X3), i.e., *A. indicus* (27.71%; 69/249), *Acinetobacter* sp002018365 (26.51%; 66/249) (an unclassified species with *Acinetobacter* sp. ANC 4845 as the reference), and *A. towneri* (12.85%; 32/249) (Table S4). Except for *A. variabilis* (11.32%; 6/53), *A. indicus* (22.64%; 12/53), *Acinetobacter* sp002018365 (20.75%; 11/53), and *A. towneri* (11.32%; 6/53) are also the predominant species carrying *tet*(X6). The distribution of *tet*(X5.2)-harboring species was similar to that of species carrying *tet*(X6), including *A. indicus* (22.64%; 12/53), *Acinetobacter* sp002018365 (20.75%; 11/53), *A. towneri* (11.32%; 6/53), *A. variabilis* (11.32%; 6/53), and *A. lwoffii* (11.32%; 6/53). These results indicate that *A. indicus* and *Acinetobacter* sp002018365 are the most prevalent species carrying tigecycline-resistant *tet*(X) genes.

To further evaluate the patterns of dissemination of *tet*(X3) and *tet*(X6) among *Acinetobacter* populations, we performed phylogenomic analysis for *tet*(X3)-/*tet*(X6)-positive isolates carried by four major species as representatives, i.e., *A. indicus*, *Acinetobacter* sp002018365, *A. towneri*, and *A. variabilis* (Fig. 4; Fig. S3). Most isolates of each species shared a distant relationship, and no epidemic clones were detected. Two interregional transmission events were detected for 4 (no SNPs) and 5 (0 or 1 SNP) isolates of *A. indicus*, and one cross-host event (pig and environment) was detected for 4 isolates (1 to 44 SNPs) of *Acinetobacter* sp002018365 (Fig. 4). The data suggested that *tet*(X3) and *tet*(X6) mainly disseminated sporadically among *Acinetobacter* populations.

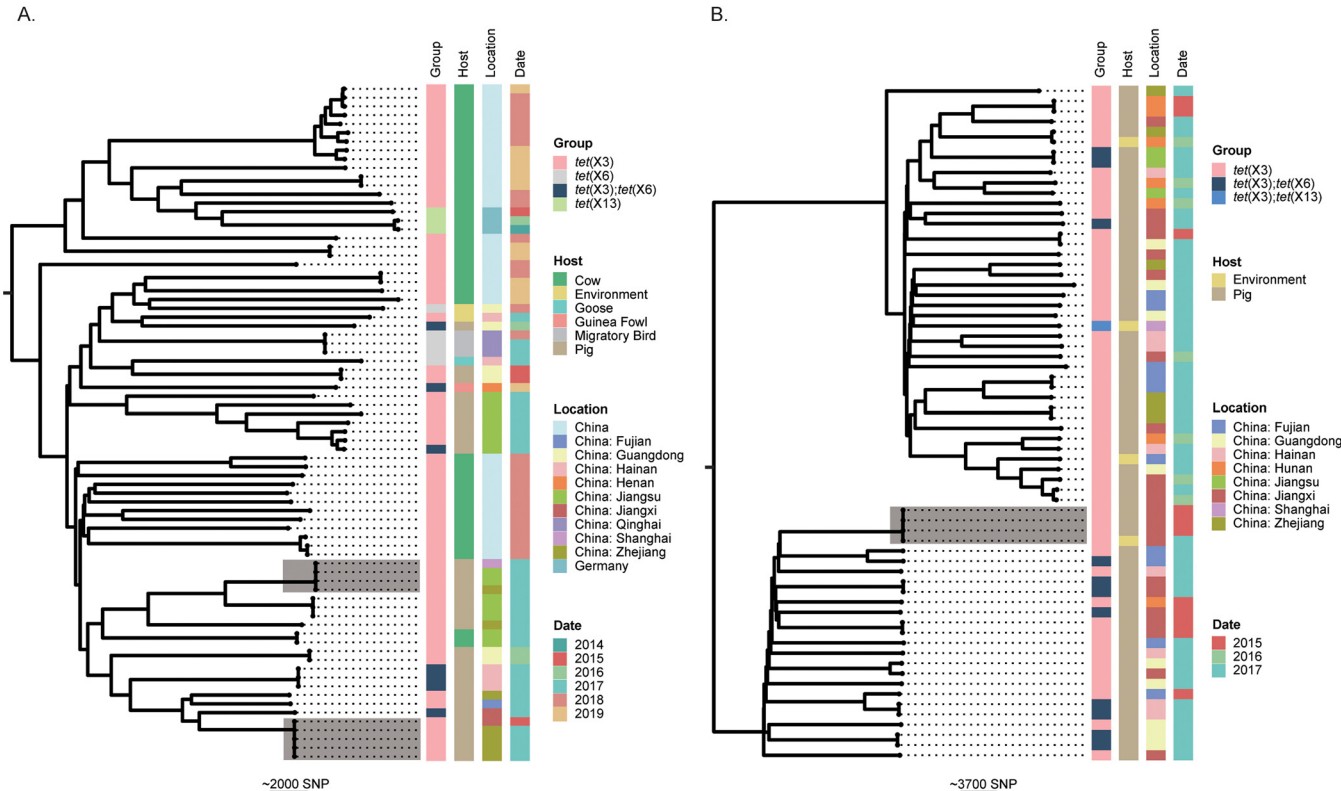

**FIG 4** Phylogenetic analysis of genomes carrying *tet*(X3)/*tet*(X6)/*tet*(X13) retrieved from GenBank. (A) The phylogenetic tree of *A. indicus*. (B) The phylogenetic tree of *Acinetobacter* sp002018365. The core-genome SNPs were used to calculate the phylogenetic trees. The trees are midpoint rooted. The *tet*(X) genes (Group), isolate source (Host), sampling location (Location), and years of isolation (Date) of strains are shown at the right side of each tree as indicated in the color keys. Two interregional transmission events for 4 and 5 strains of *A. indicus* and one cross-host event for 4 strains of *Acinetobacter* sp002018365 are highlighted by shading. The scale bar represents the number of SNPs.

**The structures of *tet*(X3)/*tet*(X6) plasmidomes are highly diverse, and no epidemic plasmids have yet been detected among *Acinetobacter*.** To explore the role of plasmids in the dissemination of *tet*(X3) and *tet*(X6) in *Acinetobacter* spp., we here intended to dissect the genetic relatedness of *tet*(X3)- and *tet*(X6)-harboring plasmids. Four circularized *tet*(X3)-/*tet*(X6)-harboring plasmids were obtained in this study, and all finished *tet*(X3)-/*tet*(X6)-harboring plasmids deposited in GenBank [*n* = 30; 18 for *tet*(X3), 6 for *tet*(X6), and 6 for *tet*(X3) and *tet*(X6)] were analyzed first. Twenty-five of the 30 publicly available plasmids were carried by *Acinetobacter* spp. Most of the 26 *tet*(X3)-harboring plasmids [including the 6 *tet*(X3)-*tet*(X6)-harboring plasmids] shared a coverage lower than 65%, indicating a highly diverse structure for the plasmidome of *tet*(X3) (Fig. 5A). Four of the 6 *tet*(X3)-*tet*(X6)-positive plasmids shared high similarity (>89.8% coverage and >85% identity), suggesting that they were derived from an ancestor. The four plasmids were hosted in *A. schindleri* and *A. indicus* strains isolated from goose and soil samples collected in different provinces of China (Fig. 5A), indicating that cross-species, cross-sector (poultry and environment), and/or cross-region transmission has occurred for these plasmids. A similar transmission event was observed for another three *tet*(X3)-encoding plasmids (pAT181, pAT184, and p10FS3-1-3) carried by *A. towneri* and a novel species of *Acinetobacter* as mentioned above (Fig. 5A).

The 5 *tet*(X6)-harboring plasmids carried by *Acinetobacter* and an unknown species share low similarities, except for pAT232 and pAT205, as mentioned above (Fig. 5B). They are different from the 3 *tet*(X6)-harboring plasmids (pAZ25, pZN3, and pZN2) carried by *Proteus* species and from the 6 *tet*(X3)-*tet*(X6)-harboring plasmids (Fig. 5B). Hence, the *tet*(X3)-*tet*(X6)-harboring plasmids might have resulted from the capture of *tet*(X6) by *tet*(X3)-harboring plasmids.

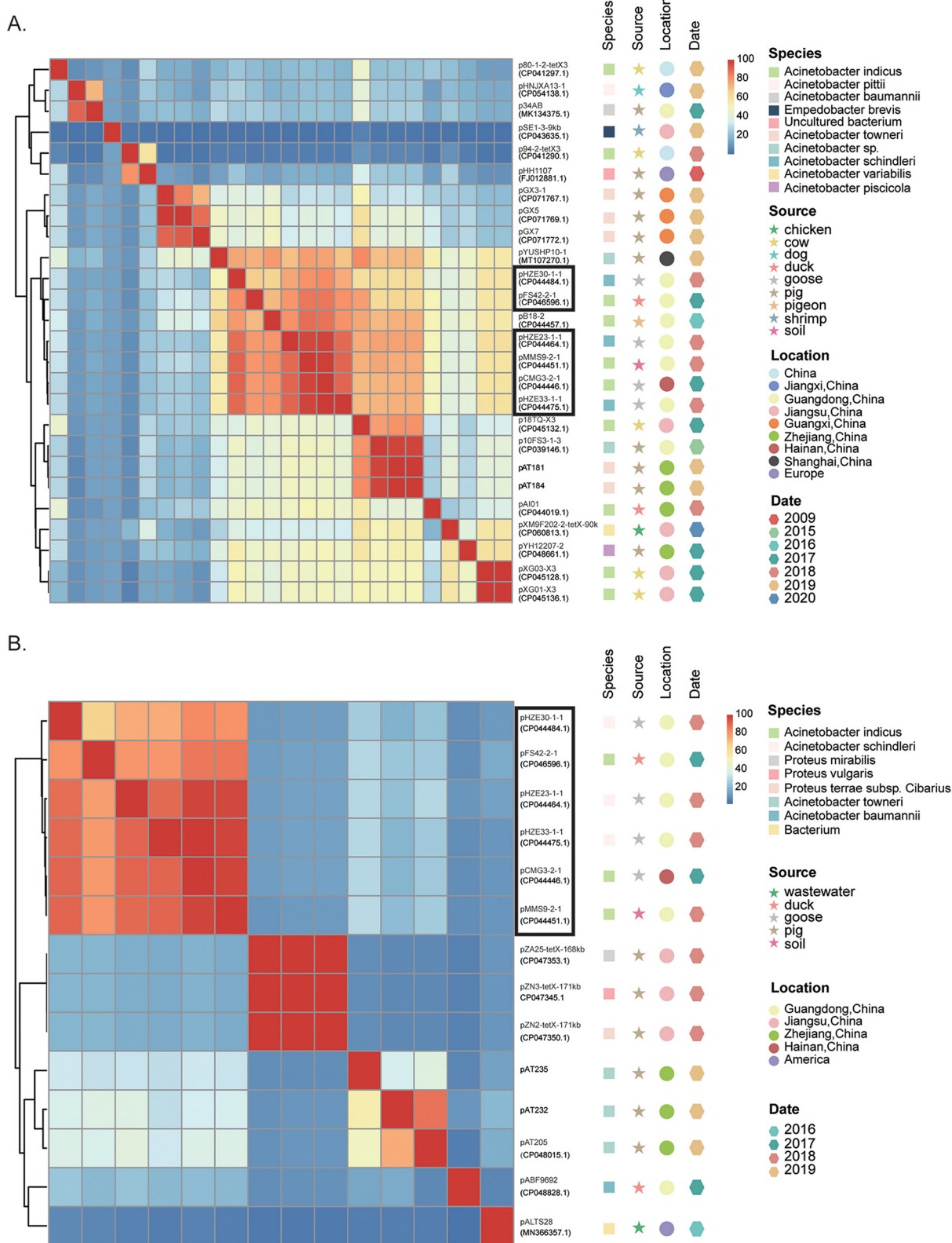

**FIG 5** Pairwise sequence comparisons between circularized *tet*(X3)-/*tet*(X6)-carrying plasmids. (A) The percentages of aligned bases between pairs of *tet*(X3)-carrying plasmids. (B) The percentages of aligned bases between pairs of *tet*(X6)-carrying plasmids. The row and column orders are the same. The host species, sampling source, sampling location, and year of isolation are shown by colored symbols to the right of the phylogenetic trees as indicated in the color keys. The six plasmids that coharbored *tet*(X3) and *tet*(X6) genes are boxed.

To further estimate the distribution of *tet*(X3)-harboring plasmids among *Acinetobacter* spp., we selected 17 plasmids out of 26 *tet*(X3)-harboring plasmids as references according to their similarities (<80% coverage and identity). The 17 plasmids were compared to the 243 *tet*(X3)-positive genomes [6 genomes with chromosome-carried *tet*(X3) were excluded] by using BLAST, and no epidemic plasmids were found (Fig. 6A). We further mapped the 243 genomic sequences against the 17 representative plasmids (Fig. 6B), and this revealed that *tet*(X3) plasmid structures were highly diverse among isolates (mean plasmid coverage range of 12.09% to 55.05%). Using a cutoff range of >80% coverage and >90% identity, we found that pGX5-like plasmids were hosted in 36 strains belonging to different species (20 *A. towneri* strains, 10 *A. variabilis* strains, 4 *Acinetobacter* sp002018365 strains, and 2 *A. indicus* strains), and p34AB-like, p94-2-*tet*X3-like, pXM9F202-2-*tet*X-90k-like, and p10FS3-1-3-like plasmids were found in 17, 9, 8, and 7 strains belonging to different species, respectively (Fig. 6A). These data suggest that the current dissemination of *tet*(X3) in *Acinetobacter* is mainly mediated by various plasmids and that cross-species transmissions mediated by a few of them might have occurred in a small proportion of cases.

## DISCUSSION

Recently identified plasmid-borne *tet*(X) genes causing the horizontal transfer of tigecycline resistance have significantly compromised the treatment effectiveness of tigecycline and, thus, have aroused considerable concern. A set of surveillance studies revealed the wide range of ecosystems in which *tet*(X) genes can be found, including soil, sewage, animals, hospitals, livestock farms, and the human gut (14–16, 19). *tet*(X)-positive isolates are especially prevalent in livestock and poultry, such as pigs, cows, and chickens, and less so in shrimp, migratory birds, and waterfowl (7, 16, 18, 19, 22, 28–30). Understanding the distribution and transmission of *tet*(X) genes in the context of One Health is imperative to efficiently control their further dissemination. In this study, we isolated *tet*(X)-positive *Acinetobacter* spp. from livestock and their surrounding environmental sources and comprehensively investigated their population structures and genetic characterizations.

According to our surveillance data, 23 *tet*(X)-positive isolates were recovered from 2 different swine farms but not from dairy farms or sheep farms. *A. towneri* was the most prevalent species carrying *tet*(X) genes in *Acinetobacter* spp., with *tet*(X3) and *tet*(X6) being the prevalent alleles (Table 1). A similar finding that *tet*(X3)-positive *Acinetobacter* species isolates were exclusively detected in intensive pig farms in China has been reported recently (20). These results suggest that the risk of dissemination of *tet*(X) genes to humans from pigs could be much higher than the risk of dissemination from other kinds of livestock.

*Acinetobacter* spp. are ubiquitous in the natural environment, and some of them, e.g., *A. baumannii*, *A. indicus*, and *A. lwoffii*, have become important opportunistic pathogens in clinical settings. Our and other studies showed that *Acinetobacter* spp. was the major reservoir of tigecycline-resistant *tet*(X) genes (17, 20, 22). Through searching *tet*(X) genes in GenBank, we found that *A. indicus*, *Acinetobacter* sp002018365, and *A. towneri* were the prevalent species carrying *tet*(X3) and *tet*(X6). Likewise, a national surveillance of *tet*(X)-positive *Acinetobacter* isolates from humans, animals, and their surrounding environments conducted between 2015 and 2018 showed that, after a novel species of *Acinetobacter*, *A. towneri* and *A. indicus* were the major hosts of *tet*(X3), *tet*(X4), and *tet*(X5) (20). Notably, most of the *tet*(X)-positive *Acinetobacter* isolates were livestock associated, raising concerns that the tigecycline-resistant *tet*(X) genes could be transmitted to humans from livestock via opportunistic pathogens of *Acinetobacter*. Our analysis showed that most of the *tet*(X)-positive *Acinetobacter* isolates disseminated sporadically; however, few interregional transmission events were detected here, highlighting the need for controlling the dissemination of *tet*(X3)- and *tet*(X6)-positive *Acinetobacter* species isolates.

Although numerous *tet*(X) genes have been continuously identified either on chromosomes or on plasmids in various bacterial species, the major vectors of tigecycline-resistant *tet*(X) genes remain unclear. Pioneering studies have shown the importance of the IS*CR2*-mediated *tet*(X) transposition structure (7, 17). The rolling-circle

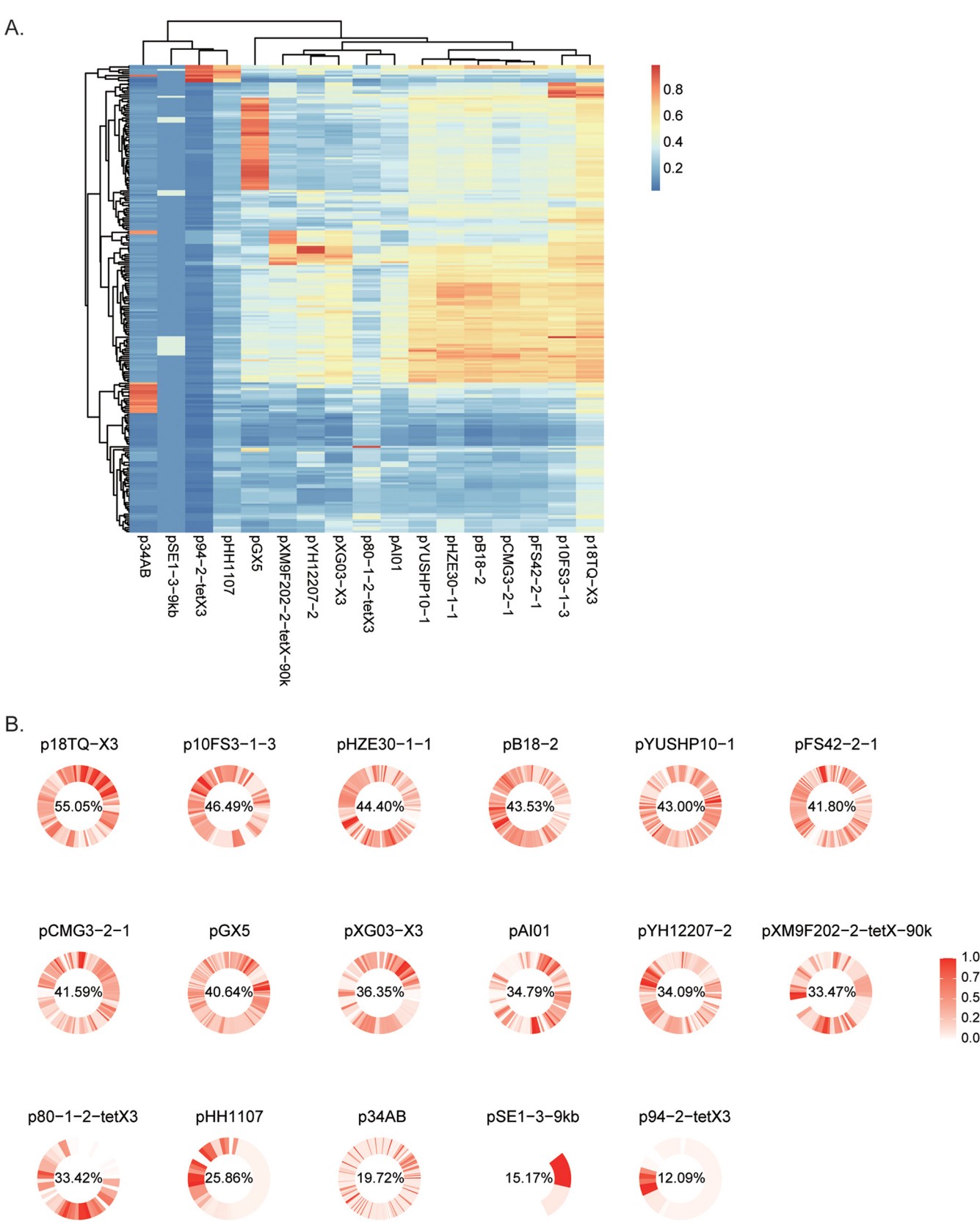

**FIG 6** Analysis of *tet*(X3) plasmidome. (A) Results for BLAST analysis of the 17 representative *tet*(X3)-carrying plasmids versus 243 *tet*(X3)-positive genomes. The heat map shows the percentages of aligned bases between pairs of *tet*(X3)-positive plasmids and genomes. (B) Conservation of reference plasmid genes among 243 genome sequences of *tet*(X3)-carrying *Acinetobacter* spp. The frequency of each gene in the reference plasmid is shown in circularized heatmaps. Genes are ordered according to the sequence of the corresponding reference plasmid. The mean coverage (%) of the reference plasmid sequence is indicated for each plasmid.

transposition has been experimentally confirmed by using the Δ*tpnF-tet*(X3)-hp-hp-IS*CR2* cassette clone, and inverse PCR assays identified IS*CR2-xerD-tet*(X3)-*res*-ORF1 and IS*CR2*-ORF2-*abh-tet*(X4) minicircles in different studies (7, 20). In our study, IS*CR2* was found upstream or downstream from *tet*(X3) and *tet*(X6) genes. Albeit we did not test the transferability of the IS*CR2*-mediated *tet*(X) transposition structure, the genetic contexts of *tet*(X3) carried by 249 genomes of *Acinetobacter* species were comprehensively compared. The proportion of the structure IS*CR2-xerD-tet*(X3)-*res*-IS*CR2* might be up to 86.35% (215/249), implying a critical role of IS*CR2* in the dissemination of *tet*(X3).

Of note, we found that a *tet*(X3)-encoding plasmid, pAT181, was self-transmissible from *A. towneri* to *A. baumannii* and conferred tetracycline resistance to the recipient. Currently, very few studies have identified self-transmissible plasmids carrying *tet*(X) genes. Chen et al. reported the conjugability of a *tet*(X3)- and *tet*(X5.3)-harboring plasmid, pYH12207-2, from *Acinetobacter piscicola* to *A. baylyi* strain ADP1 and the conjugability of a *tet*(X3)-harboring plasmid, p10FS3-1-3, from a novel *Acinetobacter* species to *A. baylyi* ADP1. However, these two plasmids did not enhance the resistance to tetracyclines in the recipient strain (20), which is different from our findings. Concerningly, pAT181, with a relatively high transfer frequency ($10^{-6}$), did not impose a fitness cost but increased the growth rate of the recipient. It is suggested that successful dissemination of resistance plasmids largely depends on the fitness cost imposed on hosts (31). No fitness cost imposed on hosts by obtaining pAT181-like plasmids would greatly facilitate their spread, and thus, might contribute to the propagation of the *tet*(X3) gene in the future. Additionally, although no epidemic plasmids carrying *tet*(X3) have been detected currently, several plasmids were found to be circulating in a small proportion of strains. These plasmids could become epidemic after transmitting to other hosts in the future.

**Conclusions.** Our study provides evidence that the predominant *tet*(X) alleles, *tet*(X3) and *tet*(X6), disseminate sporadically in *Acinetobacter* populations. Currently, the dissemination of *tet*(X3) and *tet*(X6) is mainly limited to livestock-associated sites. Continuous surveillance for *tet*(X) genes in the context of One Health is necessary to prevent them from transmitting to humans.

## MATERIALS AND METHODS

**Screenings of *tet*(X)-positive *Acinetobacter* strains.** Five hundred thirty-four nonrepetitive fecal samples were collected from 6 livestock farms located in Zhejiang Province in 2019, including 2 swine farms, 2 dairy farms, and 2 sheep farms. In addition, environmental samples were collected from soil (*n* = 72) and water (*n* = 78) surrounding the farms. All the samples were initially enriched in LB medium (5 g/liter yeast extract, 10 g/liter tryptone, 10 g/liter NaCl) for 6 h and spread on CHROMagar *Acinetobacter* medium plates (CHROMagar, Paris, France) to recover *Acinetobacter* species isolates. PCR screens of *tet*(X) alleles were performed as previously described (26).

**Antimicrobial susceptibility testing (AST).** The MICs for all the *tet*(X)-positive isolates were determined using the broth microdilution method according to the guidelines of the Clinical and Laboratory Standards Institute (CLSI) (32). The tested drugs included tigecycline, tetracycline, eravacycline, minocycline, doxycycline, demeclocycline, chlortetracycline, oxytetracycline, colistin, cefoperazone-sulbactam, trimethoprim-sulfamethoxazole, gentamicin, amikacin, levofloxacin, ciprofloxacin, meropenem, cefepime, ceftriaxone, and ceftazidime. The resistance breakpoint for tetracycline was defined as ≥16 mg/liter for *Acinetobacter* spp., *Enterobacteriaceae*, and non-*Enterobacteriaceae* according to CLSI (32). The breakpoint for tigecycline and eravacycline was delineated as >0.5 mg/liter for *Enterobacteriaceae* according to EUCAST V10 (33). *E. coli* strain ATCC 25922 was used as the quality control strain.

**WGS and bioinformatic analysis.** Genomic DNAs of the *tet*(X)-positive isolates were extracted using the Puregene yeast/bact. kit B (Qiagen, Gaithersburg, MD) according to the instructions of the manufacturer and were sequenced by using the HiSeq 4000 system (Illumina, San Diego, United States). The isolates were taxonomically assigned using GTDB-Tk (version 1.3.0) with the Genome Taxonomy Database (release 95) (34). The sequence similarities of *tet*(X)-harboring plasmids were analyzed using BRIG version 0.95 (35). Representative strains with various genetic contexts of *tet*(X) genes were selected to be further sequenced using the PromethION platform (Nanopore, Oxford, UK). Hybrid assembly of short-read and long-read sequencing data was performed using Unicycler version 0.4.8 (36).

Phylogenetic analysis was performed using Parsnp version 1.2 (37), and the numbers of single-nucleotide polymorphisms (SNPs) among the core genomes were determined by using MEGA X (38). Functional annotation was performed using the RAST server (39). Antibiotic resistance genes (ARGs) were identified using ResFinder 4.0 (40) and CARD (https://card.mcmaster.ca/) with a threshold of nucleotide identity of >90% and coverage of >90%. Synteny analysis was performed using Easyfig (41).

**Compilation of genomic data set and plasmidome analysis.** All assembled genomes of *Acinetobacter* spp. (*n* = 10,680) deposited in GenBank (as of 31 May 2021) were downloaded to search

for *tet*(X) genes. The 15 *tet*(X) alleles were queried in these genomes by BLAST comparison to their nucleotide sequences, using 99% identity and 100% coverage as the cutoff (42).

Conservation of reference plasmid genes was calculated as previously described (43). Briefly, the RedDog pipeline (https://github.com/katholt/RedDog) was used to simulate 100-bp reads from *tet*(X3)-carrying genomes. To calculate the coverage of each representative plasmid in each genome, those 100-bp reads were mapped against representative *tet*(X3)-harboring plasmids by using Bowtie2 version 2.2.9 (44). The proportion of *tet*(X3)-carrying genomes containing annotated genes of each reference plasmid was calculated according to the gene presence/absence table reported by RedDog (at least five reads covering ≥95% of the length of the gene was defined as presence), and the results were plotted as circular heatmaps using ggplot2 in R (geom_tile for heatmap grid and coord_polar for circularization).

Pairwise sequence comparison of circularized plasmids was performed as previously described (45). Briefly, the lengths of nucleotide sequences that could be aligned between pairs of plasmids and the numbers of SNPs among the aligned regions were determined by using NUCmer version 3.1 (46) from the MUMmer package. The percentages of aligned bases between pairs of complete plasmids were shown in a heatmap generated by the "gplots" package (version 3.1.1) in R version 4.0.5 (https://www.r-project.org/).

**Conjugation assay.** The transmissibility of *tet*(X3) and *tet*(X6) was evaluated by a conjugation assay. Briefly, a donor *tet*(X)-carrying *Acinetobacter* isolate (AT181) was mixed with the rifampicin-resistant *A. baumannii* strain ATCC 17978 or rifampicin-resistant *E. coli* strain EC600 as a recipient strain at a ratio of 1:1 by conjugational mating at 37°C without shaking overnight. The transconjugants were selected on LB agar plates containing rifampicin (600 mg/liter) and tigecycline (2 mg/liter). The species of all putative transconjugants were verified by using matrix-assisted laser desorption ionization–time of flight (MALDI-TOF) mass spectrometry (Hexin, Guangzhou, China). PCR verifications of *tet*(X) genes were performed for the putative transconjugants for which the species was confirmed as *A. baumannii* or *E. coli*. Transfer frequency was calculated as the number of transconjugants obtained per donor. The growth of the donor strain and transconjugants was measured by determining the optical density at 600 nm ($OD_{600}$) every 30 min. The assay was in triplicate.

**Plasmid stability testing.** Plasmid stability was estimated according to the method of a previous study with minor modifications (47). Transconjugants were cultured in antibiotic-free LB broth at 37°C for 24 h. The 24-h cultures were diluted at a ratio of 1:100 in fresh LB medium. These freshly inoculated cultures constituted time point zero, and cultures were grown at 37°C in a shaking bath (200 rpm) and serially passaged for 10 days (approximately 200 generations). Cultures were diluted and plated onto antibiotic-free LB plates every 24 h. The colonies growing on antibiotic-free LB agar plates were randomly selected (~50 colonies per day) for *tet*(X)-specific PCRs to determine the proportion of *tet*(X)-positive bacteria in each population. Plasmids were considered stable when the retention rates were still over 80% at the end of the experiment. The plasmid stability was evaluated in triplicate.

**Statistical analysis.** The unpaired *t* test was performed to compare the number of ARGs in the *tet*(X6)-carrying clone and the *tet*(X3)-carrying clone, and statistical significance was taken as a *P* value of <0.05.

**Availability of data.** The genome sequences of *tet*(X)-positive strains have been submitted to GenBank under BioProject accession number PRJNA631342, and the accession number of each genome is listed in Table 1.

## SUPPLEMENTAL MATERIAL

Supplemental material is available online only.
**SUPPLEMENTAL FILE 1**, PDF file, 6.5 MB.
**SUPPLEMENTAL FILE 2**, XLSX file, 0.3 MB.

## ACKNOWLEDGMENTS

This work was supported by the National Key Research and Development Program of China (grant number 2017YFC1200200); the National Natural Science Foundation of China (grants number 81902029 and 82172330); Shenzhen Basic Research Key projects (grant number JCYJ20200109144220704), and Shenzhen Basic Research projects (grants number JCYJ20190807144409307 and JCYJ20190807150401657).

K.Z., Y.-Y.C., Y.-H.X., and R.-C.C. designed the study. Y.-Y.C., Y.C., and F.-M.H. collected the data. Y.-Y.C. and Y.L. analyzed and interpreted the data. Y.-Y.C. and K.Z. wrote and revised the manuscript. All authors reviewed, revised, and approved the final report.

We declare no conflicts of interest.

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
