## [Reviewer comments · Microbiology Spectrum]

Microbiology Spectrum

Sporadic dissemination of *tet(X3)* and *tet(X6)* mediated by highly diverse plasmidomes among livestock-associated *Acinetobacter*

Yingying Cheng, Yang Liu, Yong Chen, Fuman Huang, Rongchang Chen, Yonghong Xiao, and Kai Zhou

Corresponding Author(s): Kai Zhou, Southern University of Science and Technology

Review Timeline:

Submission Date:	August 5, 2021
Editorial Decision:	September 6, 2021
Revision Received:	October 18, 2021
Editorial Decision:	October 19, 2021
Revision Received:	October 20, 2021
Accepted:	October 23, 2021

Editor: Tim Downing

Reviewer(s): Disclosure of reviewer identity is with reference to reviewer comments included in decision letter(s). The following individuals involved in review of your submission have agreed to reveal their identity: Miguel Angel Cevallos (Reviewer #1); Dayananda siddavattam (Reviewer #2); Nabil Karah (Reviewer #3)

Transaction Report:

DOI: <https://doi.org/10.1128/Spectrum.01141-21>

September 6, 2021

Dr. Kai Zhou
Southern University of Science and Technology
The First Affiliated Hospital
Shenzhen, Guangdong
China

Re: Spectrum01141-21 (Global dissemination of *tet(X3)* and *tet(X6)* harbored by highly diverse plasmidomes among livestock-associated *Acinetobacter*)

Dear Dr. Kai Zhou:

The reviewers have provided feedback on your paper, and I request to take their suggestions on board such that you modify the paper appropriately. Please outline in your responses how this is best achieved. On the whole the reviewers were generally positive, so paying attention to their input will help your paper.

Please pay attention to:

- Making your data publicly accessible, which is essential for publication;
- Details on the genome assembly process and evidence of quality (eg a supplementary table of metrics for each stage);
- how the strains were taxonomically assigned;
- is pAT181 a self-transmissible plasmid? As per the reviewer, homology searches for key conjugative *tra* genes will resolve this (insofar as is required for publication).
- Please ensure you correct for multiple tests in the statistical analysis (eg BH method) where you test the same dataset multiple times. Responsible scientists should report the number of tests that they apply to the same dataset. This is essential for publication.
- Your Figures are really nice, increasing the text size might help readers understand them better for the actual PDF version of the paper.

Thank you for submitting your manuscript to Microbiology Spectrum. When submitting the revised version of your paper, please provide (1) point-by-point responses to the issues raised by the reviewers as file type "Response to Reviewers," not in your cover letter, and (2) a PDF file that indicates the changes from the original submission (by highlighting or underlining the changes) as file type "Marked Up Manuscript - For Review Only". Please use this link to submit your revised manuscript - we strongly recommend that you submit your paper within the next 60 days or reach out to me. Detailed information on submitting your revised paper are below.

Link Not Available

Sincerely,

Tim Downing

Journals Department
Reviewer comments:

Reviewer #1 (Comments for the Author):

The manuscript presented by Ying-Ying Cheng and coworkers describes the dissemination of the tet(X3) and tet(X6) genes and the role of the livestock-associated *Acinetobacter* in this process. The research was correctly done, presented, in general, in a nimble way. The figure design was very good, and the conclusions of the work fit with the observations. I also think that the work is important for people interested in the epidemiology of *Acinetobacter* and especially under the perspective of "One Health". However, the work need some clarifications:

- 1_ Genome sequences are still not accessible to the public.
- 2_ The assembly statistics of the draft genomes must be presented to have a way to evaluate their quality.
- 3_ In the same way, it is important to present, in a supplementary table, the general characteristics of the close genomes.
- 4_ Authors do not describe how the strains were identified. It is very important to present the ANI analysis using the last update of *Acinetobacter* taxonomy and nomenclature (<https://apps.szu.cz/anemec/Classification.pdf>).
- 5_ A graphic of the plasmid stability will be useful.

Reviewer #2 (Comments for the Author):

The manuscript entitled "Global dissemination of tet(X3) and tet(X6) harbored by highly diverse plasmidomes among livestock-associated *Acinetobacter*" submitted by Chang et al describes dissemination of tet(X3) and tet(X6) genes among strains of *Acinetobacter* sp. isolated from the faecal samples collected from livestock establishments.

Initially the authors have identified 15 tet(X)-positive *Acinetobacter* spp. isolates and out of which 14 were *A. towneri* (n=14) and one unclassified species (n=1) (Table 1). Based on this results they concluded that *A. towneri* as prominent strain carrying tet(X) gene. Further, the authors have also claimed detection of four tet(X) alleles in the 23 isolates, including two copies of tet(X6) in one of the isolates. The authors have determined sequence of the resistant strains to understand the lateral mobility of the resistome and generated circularized plasmid sequences using long sequence reads. They have identified two tet(X3)-carrying indigenous plasmids pAT181 and pAT184 in *A. towneri* AT181 (pAT181) and *A. towneri* AT184 (pAT184). Blast analysis revealed that the sequence of pAT181 is the 100% identical to an another the tet(X3) containing plasmid, p10FS3-1-3 found in database (CP039146). The authors have also found homologues of pAT181 in *A. pittii*, *A. indicus* strains. They claimed to have nearly 326 tet(X3)-carrying plasmids in 8 *Acinetobacter* strains. However the authors have not found such similarity for tet (X6) containing plasmid backbone. Further, the authors have also conducted conjugation experiments to demonstrate the lateral mobility of the indigenous plasmid and showed lateral mobility of pAT181 into *Acinetobacter* strains but not into *E. coli* . The study is interesting and most of the inferences are drawn based on sequence information . I really don't understand the reason for conducting promoter swipings between tet(X3) and tet(X6) sequences. I see no good scientific reasons for conducting such experiments. The following are the most striking deficiencies in the manuscript.

1. It should be rewritten taking the help of a native english speaker. It is very hard to read and understand the manuscript in its current form.
 2. Lateral gene transfer takes place among *Acinetobacter* strains via multiple modes. Apart from plasmids Outer Membrane Vesicles (OMVs) play a predominant role in lateral mobility of both chromosomal encoded genes and plasmids. I don't see solid data to claim that pAT181 is a self-transmissible plasmid. Having done sequence why did they not use the sequence of circularized plasmids to identify existence of tra genes. Identifying genes coding Type 4 Secretary System (T4SS) contributing to lateral mobility of plasmids. Is pretty simple and strait forward. Simple conjugation experiments don't prove the plasmids self-transmissibility. A plasmid with oriT alone can show lateral mobility if it is present in a strain containing either chromosomally located T4SS containing plasmids or co-existing with a plasmid with a well-defined T4SS. Plasmids associated with OMV also show lateral mobility as OMVs are proven vehicles of plasmids.
- The manuscript is poorly written and it claims are primarily based on sequence results. I am not enthusiastic to recommend the manuscript for publication.

Reviewer #3 (Comments for the Author):

The study investigated the molecular epidemiology of tet(X)-positive isolates recovered from livestock samples (including some environmental samples) and the genetic context of tet(X3) and tet(X6).

The study was well designed and nicely performed. The results were interesting and relatively novel. However, the manuscript needs major revision.

Major Comments

The title should be more relevant to the study itself. "Global dissemination of" should be deleted.

The whole manuscript, especially the Results" section, should be presented clearly and directly to the point. I recommend cutting down the length of the manuscript (number of words) and deleting every unnecessary sentence.

The "discussion" section was not as good as I would expect. The authors should put more efforts to discuss the meaning of their results.

Some paragraphs need to be clearer (for example, lines 253-257) and more accurate (for example, lines 262-265)

"The phylogenetic analysis of 15 tet(X)-positive *Acinetobacter* spp. isolates showed that all but one tet(X3)-carrying *A. towneri* strains (8 out of 9) clustered together with 3-36 SNPs (Figure 1), suggesting a clonal dissemination of tet(X3) occurred in the swine farm. The other tet(X3)-carrying *A. towneri* strain AT200 clustered with the tet(X6)-carrying strains with 27,664-30,557 SNPs (Figure 1). All but two tet(X6)-positive strains showed distant relationship (26,876-31,071 SNPs), indicating that they disseminated sporadically."

I would rephrase it as following:

Eight of the tet(X3)-positive *Acinetobacter* spp. isolates clustered together with 3-36 SNPs (Figure 1), suggesting the clonal dissemination of one strain. Two of the tet(X6)-positive isolates were also clonally related (... SNPs). The remaining five isolates showed distant relationship (26,876-31,071 SNPs), indicating sporadic dissemination of these strains.

Line 78: to tet(X15) instead of (X14)

Line 563: "the expression of tet(X3) could be species-dependent" seems ambiguous and imprecise. Could the authors find out and add clearer arguments?

Authors should not mix using the present tense (for example, line 94) and past tense (for example, line 100).

Line 163: A cutoff of 99% identity and 100% coverage was used for the BLAST results. Was this cutoff subjectively selected? See for example "Hall MR, Schwarz S. Resistance gene naming and numbering: is it a new gene or not? *J Antimicrob Chemother.* 2016;71(3):569-71."

Line 189: The donors and recipients were tested and were both not able to grow on LB with rifampicin (600 mg/L) and tigecycline (2 mg/L), right? Then, if rifampicin (600 mg/L) and tigecycline (2 mg/L) are selective for the transconjugants, it is not clear for me why there was a need to confirm the species.

The authors should be careful with the terms "isolate" versus "strain". Please see Karah N, et al. Insights into the global molecular epidemiology of carbapenem non-susceptible clones of *Acinetobacter baumannii*. *Drug Resist Updat.* 2012;15(4):237-47.

Lines 331-333: What about the rep gene of pAT232 and pAT235? Do they share the same one? If yes, then they might not be different plasmids.

Minor comments

Line 70: What is "MCRs"?

Line 72: "the increasing occurrence" instead of "the recent discoveries"

Line 73: "is threatening" instead of "particularly threaten"

Line 77: "fourteen" instead of "numerous"

Line 77: "genes" instead of "alleles"

Line 79: "The Tet(X) proteins" or "The Tet(X) enzymes" instead of "These Tet(X) variants"

Line 82: Delete "Remarkably,"

Line 82/83: "the first plasmid-borne tet(X3) and tet(X4) were found in ..." instead of "the first findings of plasmid-borne tet(X3) and 83 tet(X4) identified in ..."

Line 84: "raising" instead of "raise"

Lines 88/90: Delete "However, plasmids ... been detected."

Line 92: "However" instead of "Therefore"

Line 92: "mobile elements" instead of "plasmids"

Line 94: "The tet(X) alleles have been detected in over 16 bacterial species of *Acinetobacter* spp." Instead of "Surveillance studies show that the tet(X) alleles have been detected in over 16 bacterial species with *Acinetobacter* spp."

Lines 95/96: Delete "to be the predominate one, and tet(X4) is the only allele primarily detected in *E. coli* with a low prevalence."

Line 97-100: re-write the sentences (clearer and better structured)

Line 105: "tet(X5) has so far been only" instead of "tet(X5) is solitarily"

Lines 106-108: Delete "However, it is ... sporadic dissemination."

Line 113: Delete "comprehensively"
Line 121: "In addition, environmental samples" instead of "Environmental samples"
Line 122: Delete "in parallel "
Line 122: "All the samples" instead of "These samples"
Line 129: for all the tet(X)-positive isolates
Line 135: The "resistance" breakpoint
Line 136: "defined" instead of "interpreted"
Line 138: "delineated as" instead of "interpreted as"
Line 142: of "the" tet(X)-positive
Lines 142, 145, etc.: Delete "by" Synteny
Synteny
Line 161: search "to search for tet(X) genes" instead of "to search tet(X) alleles"
Line 163: "using 99% identity and 100% coverage as a cutoff" instead of "using a cutoff as 99% identity and 100% coverage"
Line 184: which tet(X)-carrying Acinetobacter strain(s)?
Line 184: "a donor tet(X)-carrying Acinetobacter strain was" instead of "tet(X)-carrying Acinetobacter strain as a donor strain was"
Line 185: "with the rifampicin-resistant A. baumannii ATCC17978 or rifampicin-resistant E. coli EC600, as recipient strains, at the" instead of "with rifampicin-resistant A. baumannii ATCC17978 or rifampicin-resistant E. coli EC600 as a recipient strain at the"
"including 15 Acinetobacter spp. isolates (6.88%; 15/218), 5 Empedobacter stercoris, and 3 Myroides odoratimimus."
Line 245: I think it should be "6.97%, 15/215" instead of "6.88%; 15/218", right?
Line 245: No need to repeat the word "isolate" and better to go in order.
Line 247: "The Acinetobacter spp. and E. stercoris isolates were all recovered" instead of "Twenty strains were recovered"
Line 251: Delete "suggesting that A. towneri was the prevalent species carrying tet(X)s in Acinetobacter spp. population circulating at swine farms"
Lines 253-257: Re-write the results, using as clear and straightforward sentences as possible
Line 509: "their wide" instead of "the wide"
Lines 513-516: Re-write a better sentence

Staff Comments:

Preparing Revision Guidelines

Please return the manuscript within 60 days; if you cannot complete the modification within this time period, please contact me. If you do not wish to modify the manuscript and prefer to submit it to another journal, please notify me of your decision immediately so that the manuscript may be formally withdrawn from consideration by Microbiology Spectrum.

Re: Spectrum01141-21 (Global dissemination of *tet(X3)* and *tet(X6)* harbored by highly diverse plasmidomes among livestock-associated *Acinetobacter*)

Dear Dr. Kai Zhou:

The reviewers have provided feedback on your paper, and I request to take their suggestions on board such that you modify the paper appropriately. Please outline in your responses how this is best achieved. On the whole the reviewers were generally positive, so paying attention to their input will help your paper.

Please pay attention to:

- Making your data publicly accessible, which is essential for publication;

Answer: All genomes have been released.

- Details on the genome assembly process and evidence of quality (eg a supplementary table of metrics for each stage);

Answer: Thanks for the suggestions. The assembly statistics of the draft and closed genomes have been presented in Table S1.

- how the strains were taxonomically assigned;

Answer: The strains were taxonomically assigned by using gtdbtk. The information has been updated in the text and results have been listed in Table S4.

- is pAT181 a self-transmissible plasmid? As per the reviewer, homology searches for key conjugative *tra* genes will resolve this (insofar as is required for publication).

Answer: Thanks for the comments. We agree with the reviewer that identification of *tra* genes could be a simple way to predict the plasmid self-transmissibility. However, in some cases the existence of *tra* genes could not demonstrate the plasmid self-transmissibility, since some of the *tra* genes would be non-functional with unknown mechanisms. We tried to search T4SS on pAT181 according to the reviewer's suggestion, and we indeed found a *virC1* gene, which is identical to that found on a previously reported self-transmissible plasmid p10FS3-1-3 (CP039146) [Chen, C., et al., Genetic diversity and characteristics of high-level tigecycline resistance Tet(X) in *Acinetobacter* species. *Genome Med*, 2020. 12(1): p. 111.]. In addition, we performed WGS for the transconjugant ATCC17978-pAT181 and the recipient ATCC17978, and we found pAT181 in ATCC17978-pAT181 but not in ATCC17978 (Figure S2). This demonstrated that pAT181 is a self-transmissible plasmid.

- Please ensure you correct for multiple tests in the statistical analysis (eg BH method) where you test the same dataset multiple times. Responsible scientists should report the number of tests that they apply to the same dataset. This is essential for publication.

Answer: Thanks for the suggestions. We have checked the manuscript thoroughly, and the number of tests has been indicated.

- Your Figures are really nice, increasing the text size might help readers understand them better for the actual PDF version of the paper.

Answer: Thanks for the suggestions. All figures have been updated according to the suggestions.

Reviewer comments:

Reviewer #1 (Comments for the Author):

The manuscript presented by Ying-Ying Cheng and coworkers describes the dissemination of the tet(X3) and tet(X6) genes and the role of the livestock-associated *Acinetobacter* in this process. The research was correctly done, presented, in general, in a nimble way. The figure design was very good, and the conclusions of the work fit with the observations. I also think that the work is important for people interested in the epidemiology of *Acinetobacter* and especially under the perspective of "One Health". However, the work need some clarifications:

1_Genome sequences are still not accessible to the public.

Answer: All genomes sequenced in this study have been released in Genbank.

2_The assembly statistics of the draft genomes must be presented to have a way to evaluate their quality.

Answer: Thanks for the suggestions. The assembly statistics of the draft genomes have been presented in Table S1.

3_In the same way, it is important to present, in a supplementary table, the general characteristics of the close genomes.

Answer: Thanks for the suggestions. The general characteristics of the close genomes have been presented in Table S1.

4_Authors do not describe how the strains were identified. It is very important to present the ANI analysis using the last update of *Acinetobacter* taxonomy and nomenclature (<https://apps.szu.cz/anemec/Classification.pdf>).

Answer: The strains were taxonomically assigned by using gtdbtk. The information has been updated in the text and results have been listed in Table S4.

5_A graphic of the plasmid stability will be useful.

Answer: Thanks for the suggestions. Since the plasmid pAT181 was stably maintained in ATCC17978 during 10-day passage, with 100% retention rate, we think it is not necessary to show the results as a graphic.

Reviewer #2 (Comments for the Author):

The manuscript entitled "Global dissemination of tet(X3) and tet(X6) harbored by highly diverse plasmidomes among livestock-associated Acinetobacter" submitted by Chang et al describes dissemination of tet(X3) and tet(X6) genes among strains of Acinetobacter sp. isolated from the faecal samples collected from livestock establishments.

Initially the authors have identified 15 tet(X)-positive Acinetobacter spp. isolates and out of which 14 were A. towneri (n=14) and one unclassified species (n=1) (Table 1). Based on this results they concluded that A. towneri as prominent strain carrying tet(X) gene.

Further, the authors have also claimed detection of four tet(X) alleles in the 23 isolates, including two copies of tet(X6) in one of the isolates. The authors have determined sequence of the resistant strains to understand the lateral mobility of the resistome and generated circularized plasmid sequences using long sequence reads. They have identified two tet(X3)-carrying indigenous plasmids pAT181 and pAT184 in A. towneri AT181 (pAT181) and A. towneri AT184 (pAT184). Blast analysis revealed that the sequence of pAT181 is the 100% identical to an another the tet(X3) containing plasmid, p10FS3-1-3 found in database (CP039146). The authors have also found homologues of pAT181 in A. pittii, A. indicus strains. They claimed to have nearly 326 tet(X3)-carrying plasmids in 8 Acinetobacter strains. However the authors have not found such similarity for tet (X6) containing plasmid backbone. Further, the authors have also conducted conjugation experiments to demonstrate the lateral mobility of the indigenous plasmid and showed lateral mobility of pAT181 into Acinetobacter strains but not into E. coli .

The study is interesting and most of the inferences are drawn based on sequence information . I really don't understand the reason for conducting promoter swappings between tet(X3) and tet(X6) sequences. I see no good scientific reasons for conducting such experiments. The following are the most striking deficiencies in the manuscript.

Answer: Thanks for the suggestions. The results of promoter switching between tet(X3) and tet(X6) sequences have been removed in the text.

1. It should be rewritten taking the help of a native english speaker. It is very hard to read and understand the manuscript in its current form.

Answer: Thanks for the suggestions. The manuscript and language has been extensively revised.

2. Lateral gene transfer takes place among Acinetobacter strains via multiple modes. Apart from plasmids Outer Membrane Vesicles (OMVs) play a predominant role in lateral mobility of both chromosomal encoded genes and plasmids. I don't see solid data to claim that pAT181 is a self-transmissible plasmid. Having done sequence why did they not use the sequence of circularized plasmids to identify existence of tra genes. Identifying genes coding Type 4 Secretary System (T4SS) contributing to lateral mobility of plasmids. Is pretty simple and strait forward. Simple conjugation experiments don't prove the plasmids self-transmissibility. A plasmid with oriT alone can show lateral mobility if it is present in a strain containing either chromosomally located T4SS containing plasmids or co-existing with a plasmid with a well-defined T4SS. Plasmids associated with OMV also show lateral mobility as OMVs are proven vehicles of plasmids.

Answer: Thanks for the comments. We agree with the reviewer that identification of tra genes could be a simple way to predict the plasmid self-transmissibility. However, in some cases the existence of tra genes could not demonstrate the plasmid self-transmissibility, since some of the tra genes would be non-functional with unknown mechanisms. We tried to search T4SS on pAT181 according to the reviewer's suggestion, and we indeed found a virC1 gene, which is identical to that found on a previously reported self-transmissible plasmid p10FS3-1-3 (CP039146) [Chen, C., et al., Genetic diversity and characteristics of high-level tigecycline resistance Tet(X) in Acinetobacter species. Genome Med, 2020. 12(1): p. 111.]. In addition, we performed WGS for the transconjugant ATCC17978-pAT181 and the recipient ATCC17978, and we found pAT181 in ATCC17978-pAT181 but not in ATCC17978 (Figure S2). This demonstrated that pAT181 is a self-transmissible plasmid.

The manuscript is poorly written and it claims are primarily based on sequence results. I am not enthusiastic to recommend the manuscript for publication.

Answer: Thanks for the suggestions. The whole manuscript has been extensively revised and shortened.

In this study, we firstly have performed amount of laboratory work to elucidate the scientific questions, including collecting the samples, bacteria isolation, screenings of tet(X)-positive strains, Antimicrobial susceptibility testing, Conjugation assay and plasmid stability testing. Sequencing studies were further conducted to investigate the population structures and genetic characterizations of the isolates. Currently, WGS is acknowledged as an important technique to study the epidemiology of drug-resistance bacteria, since it can provide more information and ultra resolution that laboratory work cannot do. Such unreplacable advantage has been demonstrated in number of studies.

Reviewer #3 (Comments for the Author):

The study investigated the molecular epidemiology of tet(X)-positive isolates recovered from livestock samples (including some environmental samples) and the genetic context of tet(X3) and tet(X6).

The study was well designed and nicely performed. The results were interesting and relatively novel. However, the manuscript needs major revision.

Major Comments

The title should be more relevant to the study itself. "Global dissemination of" should be deleted.

Answer: Thanks for the suggestions. The title has been revised as "Sporadic dissemination of tet(X3) and tet(X6) mediated by highly diverse plasmidomes among livestock-associated Acinetobacter".

The whole manuscript, especially the Results" section, should be presented clearly and directly to the point. I recommend cutting down the length of the manuscript (number of

words) and deleting every unnecessary sentence.

Answer: Thanks for the suggestions. The whole manuscript has been extensively revised and shortened (ca. 80 lines).

The "discussion" section was not as good as I would expect. The authors should put more efforts to discuss the meaning of their results.

Answer: Thanks for the suggestions. The "discussion" section has been revised.

Some paragraphs need to be clearer (for example, lines 253-257) and more accurate (for example, lines 262-265)

Answer: Thanks for the suggestions. The sentences have been rephrased according to the reviewer's suggestions

"The phylogenetic analysis of 15 tet(X)-positive *Acinetobacter* spp. isolates showed that all but one tet(X3)-carrying *A. towneri* strains (8 out of 9) clustered together with 3-36 SNPs (Figure 1), suggesting a clonal dissemination of tet(X3) occurred in the swine farm. The other tet(X3)-carrying *A. towneri* strain AT200 clustered with the tet(X6)-carrying strains with 27,664-30,557 SNPs (Figure 1). All but two tet(X6)-positive strains showed distant relationship (26,876-31,071 SNPs), indicating that they disseminated sporadically."

I would rephrase it as following:

Eight of the tet(X3)-positive *Acinetobacter* spp. isolates clustered together with 3-36 SNPs (Figure 1), suggesting the clonal dissemination of one strain. Two of the tet(X6)-positive isolates were also clonally related (... SNPs). The remaining five isolates showed distant relationship (26,876-31,071 SNPs), indicating sporadic dissemination of these strains.

Answer: Thanks for the suggestions. The sentences have been rephrased according to the reviewer's suggestions.

Line 78: to tet(X15) instead of (X14)

Answer: Thanks for the suggestions. tet(X14) has been replaced by tet(X15), and the corresponding reference has been added.

Line 563: "the expression of tet(X3) could be species-dependent" seems ambiguous and imprecise. Could the authors find out and add clearer arguments?

Answer: Thanks for the suggestions. Indeed it would be interesting to find out the mechanism responsible for such phenotype. However, we could not figure it out in this study. Therefore this sentence is removed.

Authors should not mix using the present tense (for example, line 94) and past tense (for example, line 100).

Answer: Thanks for the suggestions. The errors have been revised.

Line 163: A cutoff of 99% identity and 100% coverage was used for the BLAST results.

Was this cutoff subjectively selected? See for example "Hall MR, Schwarz S. Resistance gene naming and numbering: is it a new gene or not? J Antimicrob Chemother. 2016;71(3):569-71."

Answer: Thanks for the comments. According to the reference suggested by the reviewer, a threshold of $\geq 2\%$ difference in the DNA or protein sequences is proposed for a new gene, and of $< 2\%$ difference at either the DNA or amino acid level or both is for variants of an existing numbered gene. In our study, we aimed to search for the 15 numbered tet(X) genes but not any new tet(X) genes in GenBank. We noted that some of the tet(X) genes shared a very high similarity of DNA sequences ($> 98\%$), e.g. tet(X) vs tet(X2) or tet(X10): 99.8%; tet(X6) vs tet(X13): 99.6%; tet(X7) vs tet(X9): 98.8%. Therefore, the use of 99% identity ($< 2\%$ difference) as a cutoff is fully consistent with the criteria proposed by the reference. The reference has been cited in the revised text.

Line 189: The donors and recipients were tested and were both not able to grow on LB with rifampicin (600 mg/L) and tigecycline (2 mg/L), right? Then, if rifampicin (600 mg/L) and tigecycline (2 mg/L) are selective for the transconjugants, it is not clear for me why there was a need to confirm the species.

Answer: Thanks for the comments. Indeed, generally only the transconjugants can be selected on the selective medium, however, in some cases, false positive donors could also be selected probably due to temporal resistance mechanisms. Therefore, we performed species confirmation to exclude the false positive donors to reduce the test size for PCR.

The authors should be careful with the terms "isolate" versus "strain". Please see Karah N, et al. Insights into the global molecular epidemiology of carbapenem non-susceptible clones of *Acinetobacter baumannii*. Drug Resist Updat. 2012;15(4):237-47.

Answer: Thanks for the suggestions. According to the explanation in the reference, "isolate" is more suitable than "strain" in our manuscript. The description has been revised in the text.

Lines 331-333: What about the rep gene of pAT232 and pAT235? Do they share the same one? If yes, then they might not be different plasmids.

Answer: Thanks for the comments. We analyzed the rep gene of pAT232 and pAT235, and the sequence is identical. Therefore the sequence has been rephrased as "The two tet(X6)-harboring circularized plasmids pAT232 and pAT235 shared as low as 38% coverage and 99.95% identity, but the sequences of their rep genes were identical, indicating that they might originate from a common ancestor."

Minor comments

Line 70: What is "MCRs"?

Answer: Apologize for this error. "MCRs" is the abbreviation of mobilized colistin resistance. "MCRs" has been replaced by "mobilized colistin resistance (mcr) genes" in the text.

Line 72: "the increasing occurrence" instead of "the recent discoveries"

Answer: Thanks for the suggestion. "the recent discoveries" has been replaced by "the increasing occurrence".

Line 73: "is threatening" instead of "particularly threaten"

Answer: Thanks for the suggestion. "particularly threaten" has been replaced by "is threatening".

Line 77: "fourteen" instead of "numerous"

Answer: Thanks for the suggestion. "numerous" has been replaced by "fourteen".

Line 77: "genes" instead of "alleles"

Answer: Thanks for the suggestion. "alleles" has been replaced by "genes".

Line 79: "The Tet(X) proteins" or "The Tet(X) enzymes" instead of "These Tet(X) variants"

Answer: Thanks for the suggestion. "These Tet(X) variants" has been replaced by "The Tet(X) enzymes".

Line 82: Delete "Remarkably,"

Answer: Thanks for the suggestion. "Remarkably," has been deleted.

Line 82/83: "the first plasmid-borne tet(X3) and tet(X4) were found in ..." instead of "the first findings of plasmid-borne tet(X3) and 83 tet(X4) identified in ..."

Answer: Thanks for the suggestion. "the first findings of plasmid-borne tet(X3) and 83 tet(X4) identified in ..." has been replaced by "the first plasmid-borne tet(X3) and tet(X4) were found in ...".

Line 84: "raising" instead of "raise"

Answer: Thanks for the suggestion. "raise" has been replaced by "raising".

Lines 88/90: Delete "However, plasmids ... been detected."

Answer: Thanks for the suggestion. "However, plasmids ... been detected." has been deleted.

Line 92: "However" instead of "Therefore"

Answer: Thanks for the suggestion. "Therefore" has been replaced by "However".

Line 92: "mobile elements" instead of "plasmids"

Answer: Thanks for the suggestion. "plasmids" has been replaced by "mobile elements".

Line 94: "The tet(X) alleles have been detected in over 16 bacterial species of Acinetobacter spp." Instead of "Surveillance studies show that the tet(X) alleles have been detected in over 16 bacterial species with Acinetobacter spp."

Answer: Thanks for the suggestion. Since the 16 species do not all belong to *Acinetobacter* spp., we therefore revised this sentence as "The tet(X) genes have been detected in over 16 bacterial species, and *Acinetobacter* spp. is one of major hosts".

Lines 95/96: Delete "to be the predominate one, and tet(X4) is the only allele primarily detected in *E. coli* with a low prevalence."

Answer: Thanks for the suggestion. "to be the predominate one, and tet(X4) is the only allele primarily detected in *E. coli* with a low prevalence." has been deleted.

Line 97-100: re-write the sentences (clearer and better structured)

Answer: Thanks for the suggestion. The sentences have been fully restructured.

Line 105: "tet(X5) has so far been only" instead of "tet(X5) is solitarily"

Answer: Thanks for the suggestion. "tet(X5) is solitarily" has been replaced by "tet(X5) has so far been only".

Lines 106-108: Delete "However, it is ... sporadic dissemination."

Answer: Thanks for the suggestion. "However, it is ... sporadic dissemination." has been deleted.

Line 113: Delete "comprehensively"

Answer: Thanks for the suggestion. "comprehensively" has been deleted.

Line 121: "In addition, environmental samples" instead of "Environmental samples"

Answer: Thanks for the suggestion. "Environmental samples" has been replaced by "In addition, environmental samples".

Line 122: Delete "in parallel "

Answer: Thanks for the suggestion. "in parallel " has been deleted.

Line 122: "All the samples" instead of "These samples"

Answer: Thanks for the suggestion. "These samples" has been replaced by "All the samples".

Line 129: for all the tet(X)-positive isolates

Answer: Thanks for the suggestion. It has been revised.

Line 135: The "resistance" breakpoint

Answer: Thanks for the suggestion. It has been revised.

Line 136: "defined" instead of "interpreted"

Answer: Thanks for the suggestion. "interpreted" has been replaced by "defined".

Line 138: "delineated as" instead of "interpreted as"

Answer: Thanks for the suggestion. "interpreted as" has been replaced by "delineated as".

Line 142: of "the" tet(X)-positive

Answer: Thanks for the suggestion. "the" has been added.

Lines 142, 145, etc.: Delete "by" Synteny

Answer: Thanks for the suggestion. This has been revised through the text.

Line 161: search "to search for tet(X) genes" instead of "to search tet(X) alleles"

Answer: Thanks for the suggestion. " instead of "to search tet(X) alleles" has been replaced by "to search for tet(X) genes".

Line 163: "using 99% identity and 100% coverage as a cutoff" instead of "using a cutoff as 99% identity and 100% coverage"

Answer: Thanks for the suggestion. "using a cutoff as 99% identity and 100% coverage" has been replaced by "using 99% identity and 100% coverage as a cutoff".

Line 184: which tet(X)-carrying Acinetobacter strain(s)?

Answer: Thanks for the suggestion. The isolate ID AT181 has been added.

Line 184: "a donor tet(X)-carrying Acinetobacter strain was" instead of "tet(X)-carrying Acinetobacter strain as a donor strain was"

Answer: Thanks for the suggestion. "tet(X)-carrying Acinetobacter strain as a donor strain was" has been replaced by "a donor tet(X)-carrying Acinetobacter strain was".

Line 185: "with the rifampicin-resistant A. baumannii ATCC17978 or rifampicin-resistant E. coli EC600, as recipient strains, at the" instead of "with rifampicin-resistant A. baumannii ATCC17978 or rifampicin-resistant E. coli EC600 as a recipient strain at the"

Answer: Thanks for the suggestion. "with rifampicin-resistant A. baumannii ATCC17978 or rifampicin-resistant E. coli EC600 as a recipient strain at the" has been replaced by "with the rifampicin-resistant A. baumannii ATCC17978 or rifampicin-resistant E. coli EC600, as recipient strains, at the".

"including 15 Acinetobacter spp. isolates (6.88%; 15/218), 5 Empedobacter stercoris, and 3 Myroides odoratimimus."

Line 245: I think it should be "6.97%, 15/215" instead of "6.88%; 15/218", right?

Answer: Apologize for this error. The data has been revised.

Line 245: No need to repeat the word "isolate" and better to go in order.

Answer: Thanks for the suggestion. "isolate" has been deleted.

Line 247: "The Acinetobacter spp. and E. stercoris isolates were all recovered" instead of "Twenty strains were recovered"

Answer: Thanks for the suggestion. "Twenty strains were recovered" has been replaced by "The Acinetobacter spp. and E. stercoris isolates were all recovered".

Line 251: Delete "suggesting that A. townneri was the prevalent species carrying tet(X)s in Acinetobacter spp. population circulating at swine farms"

Answer: Thanks for the suggestion. The sentence has been deleted.

Lines 253-257: Re-write the results, using as clear and straightforward sentences as possible

Answer: Thanks for the suggestion. This part has been restructured.

Line 509: "their wide" instead of "the wide"

Answer: Thanks for the suggestion. "the wide" has been replaced by "their wide".

Lines 513-516: Re-write a better sentence

Answer: Thanks for the suggestion. The sentence has been rewritten.

October 19, 2021

Dr. Kai Zhou
Southern University of Science and Technology
The First Affiliated Hospital
Shenzhen, Guangdong
China

Re: Spectrum01141-21R1 (Sporadic dissemination of *tet(X3)* and *tet(X6)* mediated by highly diverse plasmidomes among livestock-associated *Acinetobacter*)

Dear Dr. Kai Zhou:

Thank you for making those changes. Most are fine. Two items left:

[1] As outlined before, please ensure you correct for multiple tests in the statistical analysis (eg Benjamini-Hochberg, etc) because you tested the same dataset multiple times. You still have: "Statistical analysis was performed using unpaired t-test analysis, and statistical significance is taken as $p < 0.05$ " - this is not proper reporting. You need to outline exactly what you did and where it is related to the later parts of the paper. This is essential for publication.

[2] Please add the correct BioProject ID to your study PRJNA631342. The IDs you supplied are cumbersome for readers to access, it is easier to access the whole set, and so the BioProject ID would be good to mention in the paper.

Thank you for submitting your manuscript to Microbiology Spectrum. When submitting the revised version of your paper, please provide (1) point-by-point responses to the issues raised by the reviewers as file type "Response to Reviewers," not in your cover letter, and (2) a PDF file that indicates the changes from the original submission (by highlighting or underlining the changes) as file type "Marked Up Manuscript - For Review Only". Please use this link to submit your revised manuscript - we strongly recommend that you submit your paper within the next 60 days or reach out to me. Detailed information on submitting your revised paper are below.

Link Not Available

Sincerely,

Tim Downing

Journals Department
Reviewer comments:

Staff Comments:

Preparing Revision Guidelines

To submit your modified manuscript, log onto the eJP submission site at <https://spectrum.msubmit.net/cgi-bin/main.plex>. Go to

Author Tasks and click the appropriate manuscript title to begin the revision process. The information that you entered when you first submitted the paper will be displayed. Please update the information as necessary. Here are a few examples of required updates that authors must address:

Please return the manuscript within 60 days; if you cannot complete the modification within this time period, please contact me. If you do not wish to modify the manuscript and prefer to submit it to another journal, please notify me of your decision immediately so that the manuscript may be formally withdrawn from consideration by Microbiology Spectrum.

[1] As outlined before, please ensure you correct for multiple tests in the statistical analysis (eg Benjamini-Hochberg, etc) because you tested the same dataset multiple times. You still have: "Statistical analysis was performed using unpaired t-test analysis, and statistical significance is taken as $p < 0.05$ " - this is not proper reporting. You need to outline exactly what you did and where it is related to the later parts of the paper. This is essential for publication.

Answer: Thanks for the comments. The statistical analysis was only performed to compare the number of ARGs in the *tet(X6)*-carrying clone and the *tet(X3)*-carrying clone using the unpaired *t*-test analysis, therefore, we revised the sentence as "The unpaired *t*-test analysis was performed to compare the number of ARGs in the *tet(X6)*-carrying clone and the *tet(X3)*-carrying clone, and statistical significance is taken as $p < 0.05$ ".

[2] Please add the correct BioProject ID to your study PRJNA631342. The IDs you supplied are cumbersome for readers to access, it is easier to access the whole set, and so the BioProject ID would be good to mention in the paper.

Answer: Thanks for the suggestions. The BioProject ID has been added to the text, and the "Availability of data and materials" was revised as "The genome sequences of *tet(X)* positive strains have been submitted to GenBank under BioProject accession number PRJNA631342, and the accession number of each genome is listed in Table 1."

October 23, 2021

Dr. Kai Zhou
Southern University of Science and Technology
The First Affiliated Hospital
Shenzhen, Guangdong
China

Re: Spectrum01141-21R2 (Sporadic dissemination of *tet(X3)* and *tet(X6)* mediated by highly diverse plasmidomes among livestock-associated *Acinetobacter*)

Dear Dr. Kai Zhou:

Your manuscript has been accepted, and I am forwarding it to the ASM Journals Department for publication. You will be notified when your proofs are ready to be viewed.

Sincerely,

Tim Downing
Editor, Microbiology Spectrum

Journals Department
Supplemental Material: Accept
Supplemental Table 4: Accept